# Mechanisms underlying pathological cortical bursts during metabolic depletion

Shrey Dutta [1,2,3] ✉, Kartik K. Iyer [1], Sampsa Vanhatalo [4], Michael Breakspear[3,5] & James A. Roberts [1,2]

Cortical activity depends upon a continuous supply of oxygen and other metabolic resources. Perinatal disruption of oxygen availability is a common clinical scenario in neonatal intensive care units, and a leading cause of lifelong disability. Pathological patterns of brain activity including burst suppression and seizures are a hallmark of the recovery period, yet the mechanisms by which these patterns arise remain poorly understood. Here, we use computational modeling of coupled metabolic-neuronal activity to explore the mechanisms by which oxygen depletion generates pathological brain activity. We find that restricting oxygen supply drives transitions from normal activity to several pathological activity patterns (isoelectric, burst suppression, and seizures), depending on the potassium supply. Trajectories through parameter space track key features of clinical electrophysiology recordings and reveal how infants with good recovery outcomes track toward normal parameter values, whereas the parameter values for infants with poor outcomes dwell around the pathological values. These findings open avenues for studying and monitoring the metabolically challenged infant brain, and deepen our understanding of the link between neuronal and metabolic activity.

The brain is an energy-demanding organ whose activity depends upon a rich supply of metabolic resources, including oxygen and glucose. Compromised supply of oxygen and other critical resources is a central factor in many neurological disorders such as epilepsy, movement disorders, and dementia[1]. Neurological complications that are common consequences of stroke, cardiac arrest, asphyxia are also likely caused by metabolic disturbances[2–4]. Compromised oxygen supply is a common perinatal insult, with crucial consequences for neurodevelopment[5,6]. Yet, the dynamic interdependence of cortical activity and metabolic supply—and the ways in which this can be disrupted—are not well understood.

Scalp electroencephalography (EEG) is routinely used in the neonatal intensive care unit (NICU), with monitoring of pathological patterns such as seizures and burst suppression (BS)[7] a part of standard care. The hallmark of BS during recovery from acute oxygen deprivation is high amplitude bursts of activity separated by quiescent periods. Neonatal BS exhibits scale-free dynamics underlying highly variable amplitudes. Bursts initially have asymmetric shapes that become more symmetric as recovery progresses[8]. Another form of BS is seen in adults under anesthesia, where cortical bursts have relatively constant amplitude[9]. Modeling of anesthesia-induced BS has identified a mechanism of fast-slow bursting driven by metabolic resource depletion[9,10]. In contrast, the mechanisms of neonatal post ischemic-hypoxic BS remain poorly explored.

Despite the high clinical relevance of metabolic insults, models of brain network activity have largely focused solely on neural activity. But neurons do not exist in a vacuum: their activity is intertwined with the metabolic, synaptic, and ionic resources available to them. The main energy expenditure for signaling is the active transport of intra- and extracellular ions to maintain the resting membrane potential[11–13].

[1]Brain Modelling Group, QIMR Berghofer Medical Research Institute, Brisbane, QLD, Australia. [2]Faculty of Medicine, University of Queensland, Brisbane, QLD, Australia. [3]School of Psychological Sciences, College of Engineering, Science and Environment, University of Newcastle, Callaghan, NSW, Australia. [4]Pediatric Research Center, Department of Physiology, Helsinki University Hospital, University of Helsinki, Helsinki, Finland. [5]School of Medicine and Public Health, College of Health and Medicine, University of Newcastle, Callaghan, NSW, Australia. ✉e-mail: s.dutta@uq.net.au

Neurons source energy from hydrolysis of adenosine 5′-triphosphate (ATP), primarily derived from aerobic oxidation of glucose[14,15]. Perturbations of oxygen availability lead to cascades of cellular-level changes in neuronal function. Despite knowledge of these cellular processes, their propagation to the scales of larger networks is not understood.

Here, we use clinical neurophysiological recordings to study the interdependence of neural activity and its metabolic resource pool. We first develop a network model of neurons coupled to their available oxygen and ionic resources. We next simulate conditions of hypoxia, and analyze the ensuing neuronal dynamics. We then examine trajectories through parameter space that correspond to different recovery scenarios, and validate the model against empirical clinical EEG data.

## Results
### Modeling metabolically constrained cortical dynamics
Local field potentials reflect activity in local neural populations. We hence model a local neuronal circuit composed of 400 modified Hodgkin-Huxley neurons coupled to their metabolic supply[16,17] (Fig. 1a; see Methods). Neuronal activity is coupled to $O_2$ dynamics via $Na^+$-$K^+$ pumps, reflecting the fact that most of the brain's energy expenditure for signaling is associated with maintaining the functioning of $Na^+$-$K^+$ pumps[12,13]. Each neuron receives input from ~80 synaptically coupled neurons (~64 excitatory and ~16 inhibitory) through local random connectivity (see Methods). This random network architecture has been shown to generate self-sustaining activity[18–20]. Therefore, we do not include a (deterministic or stochastic) external drive.

We investigate the parameter space of external potassium and oxygen supply to understand the effect of metabolism on neural activity. The oxygen supply from the blood vessels is modeled as diffusion from a reservoir of concentration $[O_2]_{Buffer}$[16]. Similarly, the diffusion of potassium into the extracellular space from blood vessels and other sources is modeled by the concentration ($[K^+]_{Buffer}$) of potassium in the reservoir[16,21]. Previous research has identified $[O_2]_{Buffer}$ and $[K^+]_{Buffer}$ as important control parameters for single-neuron dynamics and experimental preparations[16,17,22].

The activity of the energy-consuming $Na^+$-$K^+$ pumps (Eq. (4)) is influenced by several factors (Fig. 1i). The extracellular potassium ($[K^+]_o$), intracellular sodium ($[Na^+]_i$), and extracellular oxygen ($[O_2]_o$) concentrations directly impact the pump currents (Eqs. (1–3)). Meanwhile, $[O_2]_{Buffer}$ and $[K^+]_{Buffer}$ set the baseline values for $[O_2]_o$ (Eq. (4)) and $[K^+]_o$ (Eq. (5)), respectively. As a result, an increase in $[K^+]_{Buffer}$ indirectly increases the baseline energy demand by raising the equilibrium value of $[K^+]_o$.

To benchmark our model of the resource-depleted cortex, we analyzed scalp EEG acquired in routine clinical care (Fig. 1b) from 17 infants following birth asphyxia. Modeled cortical dynamics simulated for metabolically challenged hypoxic conditions (Fig. 1c) were compared to physiological metrics derived from infant EEG exhibiting pathological activity during recovery from asphyxia (Fig. 1d–f)[8,23]. We mapped the emergent dynamical regimes in the parameter space of $[K^+]_{Buffer}$ and $[O_2]_{Buffer}$ (Fig. 1g), and through parameter estimation, identified trajectories of model parameters corresponding to the EEG time series (Fig. 1h).

### Scale-free burst suppression
Burst suppression activity in infants following ischemic-hypoxic injury exhibits stereotypical, high amplitude bursts of highly variable sizes interspersed with periods of low EEG activity (Fig. 2a–d; zoomed 5 min windows in Fig. 2e–h). We quantified these bursty EEG patterns using burst metrics sensitive to scale-free dynamics. It has previously been shown that these data exhibit approximately scale-free distributions of burst areas and durations, and that the average shape of these bursts is asymmetric[8]. Here, we fitted strictly truncated power law distributions (see Methods) to burst area distributions (Fig. 2i–l) and burst duration

distributions (Fig. 2m–p). We calculated the number of orders of magnitude (O) and scaling exponent (E) of each fit. We found that even short 5 min windows exhibit scale-free distributions over several orders of magnitude. The scale-free nature of these bursts enables the averaging together of bursts to seek a common underlying shape[24]. This revealed asymmetric burst shapes, which can be quantified by calculating the burst asymmetry (Σ) and burst sharpness (K) (Fig. 2q–t). These six metrics provide clinically relevant, quantitative comparators between NICU-monitored EEG and model-derived simulations.

In the next section, we explore the types of dynamics the model generates. Then, we infer the model's physiological parameters and their trajectories through parameter space reflecting the progression towards continuous EEG in infants recovering from birth asphyxia.

### Physiological regime: self-sustaining asynchronous irregular (AI) activity
We explored the model's activity during healthy and pathological values of metabolic parameters, starting with the healthy (physiological) case. Randomly connected neuronal networks have been shown to generate self-sustaining asynchronous (i.e., pairwise cross-correlation CC < 0.1) and irregular (i.e., coefficient of variation of the inter-spike interval $CV_{ISI}$ > 1) activity[18–20], similar to cortical activity of awake cats, monkeys, and humans[18,19,25]. Therefore, we consider asynchronous irregular (AI) activity as physiologically healthy. We first examined the case of adequate metabolic supply and sought a network regime corresponding to AI activity.

Setting $[O_2]_{Buffer}$ to a physiological value of 32 mg/L[16,17], and $[K^+]_{Buffer}$ to its physiological value of 3.5 mM[17], yields self-sustaining AI activity states, when the maximum synaptic conductances are inhibition dominated (maximum excitatory synaptic conductance $G_{ex}$ = 0.022 mS/cm², and maximum inhibitory synaptic conductance $G_{inh}$ = 0.374 mS/cm²), consistent with prior work[18–20]. The maximum synaptic conductances are fixed to these values for the remainder of the paper.

With this network connectivity, the model generates self-sustaining AI dynamics (with $CV_{ISI}$ = 1.01 and CC = 0.04, computed from 60 s simulation) without any external source of noise or external current (Fig. 3a). The only source of randomness comes from the static random connectivity, also termed quenched noise[19]. The average instantaneous firing rate time series ($\phi^{fr}(t)$; see Methods) and the average of post-synaptic currents of the excitatory neurons ($\Phi^{syn}(t)$; see Methods) are widely used as estimates of local field potentials[26], representing population-level activity of a network of neurons. The asynchronous and irregular nature of the dynamics can also be seen in $\phi^{fr}(t)$ (Fig. 3b), and $\phi^{syn}(t)$ (Fig. 3c). While spiking activity is irregular, it nevertheless exhibits beta-band oscillations with a broad spectral peak around 31 Hz, reflecting a periodic modulation of the sporadic spike rate (Fig. 3d).

This network rhythm emerges through the interaction between excitatory and inhibitory neurons. Increasing the time constant of inhibitory synapses slows down the response of the inhibitory neurons to incoming input. This delay allows excitatory neurons to increase their firing rate transiently before inhibition reduces it again, which drives a roughly oscillatory modulation of the network firing rate. The peak frequency of ~31 Hz varies roughly linearly with $\tau_{inh}$ in the vicinity of our nominal parameter set (Supplementary Fig. S1).

Balanced excitatory and inhibitory activity is an indicator of a healthy brain state, as observed experimentally in vitro[27], in vivo[28], and in human electrophysiological data[25]. Deviations from this balance are an important marker of pathological states[25]. We estimate E-I balance by calculating the logarithm of the ratio of the mean excitatory post-synaptic current (EPSC) and mean inhibitory post-synaptic current (IPSC) of excitatory and inhibitory neurons. An E-I balance of 0 indicates a perfect balance between EPSC and IPSC; positive values indicate more excitation and negative values indicate more inhibition. Within the AI state, we observe that the E-I balance is maintained close

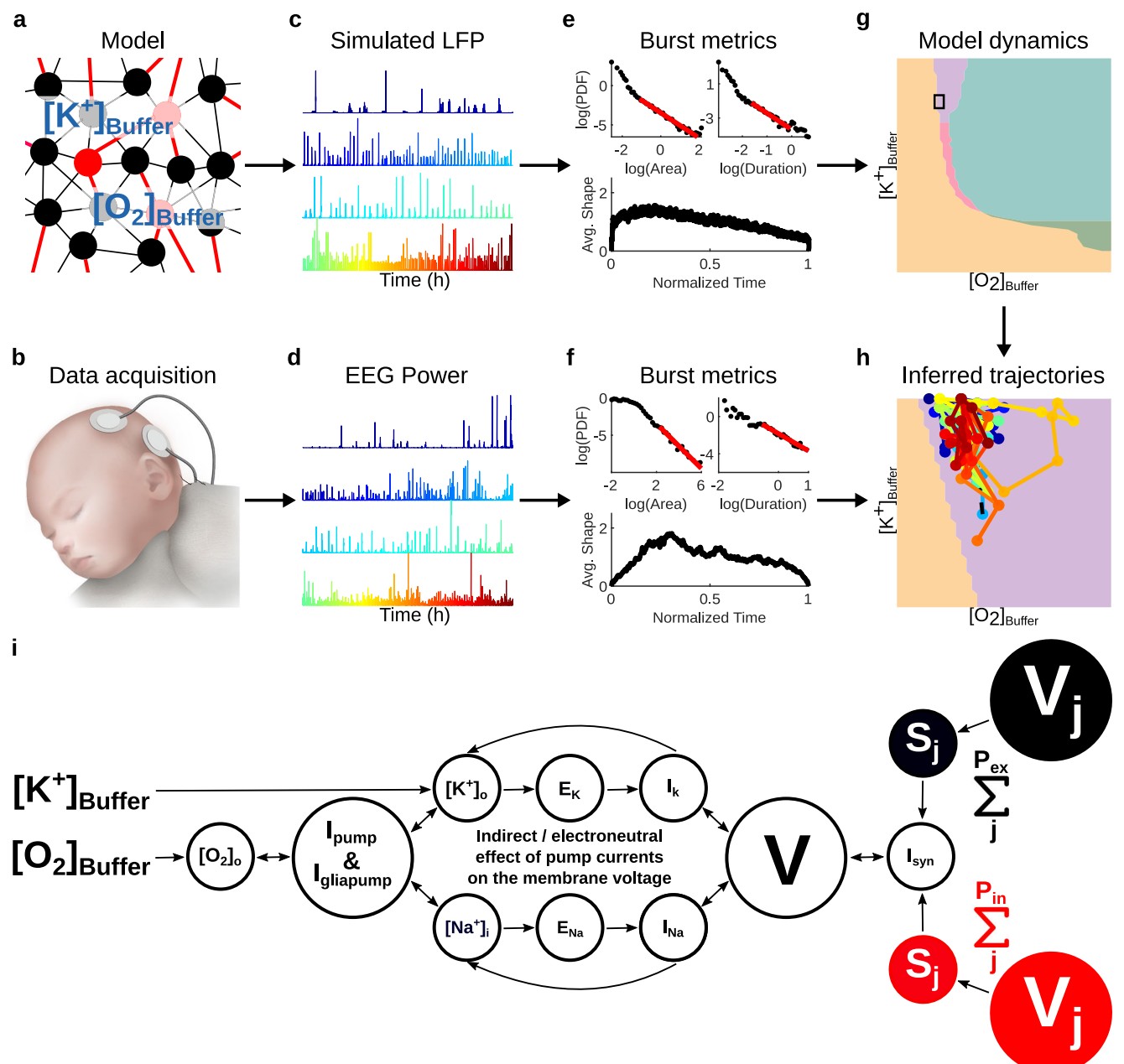

**Fig. 1 | Overview of the analysis. a** Local brain activity is modeled with a network of 400 modified Hodgkin-Huxley neurons (320 excitatory in black and 80 inhibitory in red) with $O_2$ dynamics. $O_2$ and $K^+$ diffuse into the extracellular space from reservoirs with concentrations $[O_2]_{Buffer}$ and $[K^+]_{Buffer}$. **b** Two channel, biparietal electroencephalogram (EEG) was recorded from 17 infants during recovery from ischemic-hypoxic insults at birth. **c** Simulated local field potentials (LFP) under hypoxic conditions. **d** Infant EEG instantaneous power exhibiting burst suppression. **e, f** Six measures of burst statistics were estimated from the EEG and simulated time series (see Fig. 2 for details): orders of magnitude and exponents of both the distribution of burst area and the distribution of duration (total 4 statistics); and asymmetry and sharpness of average burst-shapes from duration 1280 ms to 5120 ms (total 2 statistics). **g** We systematically mapped the emergent dynamical regimes (denoted by colors) in the parameter space of $[K^+]_{Buffer}$ and $[O_2]_{Buffer}$. **h** By triangulating the infant EEG metrics within the corresponding model parameter space,

we inferred likely parameter trajectories of individual infants. **i** Model schematics. We use a network of 400 neurons (320 excitatory and 80 inhibitory) such that each neuron receives synaptic inputs ($S_j$) from 80 random neurons. The dendritic summation of these inputs results in the postsynaptic current ($I_{syn}$) for the neuron. $I_{syn}$ modulates the neuron's membrane potential ($V$). $V$ is also modulated by the intrinsic ion currents ($I_K$ and $I_{Na}$), which result from the net ion flow between the intracellular and extracellular spaces. Intracellular and extracellular ion concentrations ($[K^+]$ and $[Na^+]$) establish gradients across the membrane (reversal potentials $E_K$ and $E_{Na}$). Ion pumps ($I_{pump}$ and $I_{gliapump}$) modulate ion concentrations to maintain concentration gradients, expending energy derived from $O_2$ bonds[81]. The extracellular concentration of $O_2$ is mediated by $[O_2]_{Buffer}$. The model also incorporates diffusion of potassium from distal sources parameterized by $[K^+]_{Buffer}$. Panel (**b**) illustration by Madeleine Kersting Flynn.

to 0 indicating healthy dynamics (Fig. 3e). Small fluctuations away from zero are common to both populations, yielding a zero reverting effect.

The metabolically well-resourced state in our model thus exhibits statistics consistent with experimentally observed self-sustaining AI

states. In the subsequent results, we keep the synaptic strengths ($G_{ex}$ and $G_{inh}$) fixed to the values of the self-sustaining AI activity state.

**AI regime co-exists with an isoelectric regime.** An isoelectric state with no firing activity exists when $[K^+]_{Buffer}$ or $[O_2]_{Buffer}$ are very low.

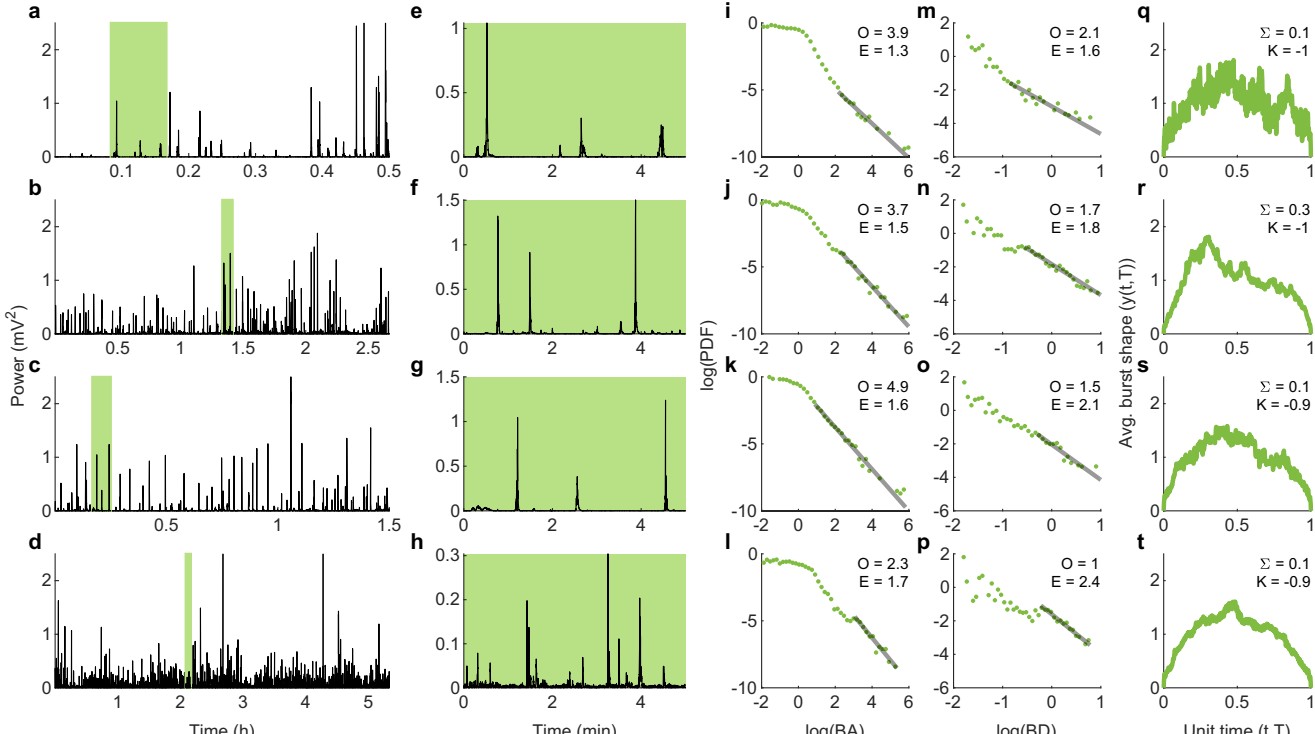

**Fig. 2 | Exemplar EEG time series of a neonate recovering from birth asphyxia.** **a**–**d** Four different epochs of BS recorded sequentially within the first 14 h post birth. The density of bursts increases with epochs, consistent with a progression towards continuous EEG. Green rectangles mark the 5 min windows analyzed in the subsequent panels. A small number of the largest bursts in panels a and d are truncated for clarity. **e**–**h** Zooms of the green windows shown in (**a**–**d**). **i**–**l** Burst area probability densities for the bursts extracted from the time series in (**e**–**h**).

Logarithmically binned probability density functions (PDFs) of burst area (green) with maximum likelihood fits to strictly truncated power-law distributions (black). The orders of magnitude (O) and the exponent (E) of the fit are displayed in each figure panel. **m**–**p** Burst duration probability densities (green) for the bursts extracted from the time series in (**e**–**h**), with strictly truncated power-law fits (black). **q**–**t** Average shape of bursts ($y(t,T)$) of duration from 1280 s to 5120 s. The estimates of asymmetry ($\Sigma$) and sharpness ($K$) are shown as insets.

This state is similar to the clinically observed iso-electric state having negligible neuronal activity, classified by clinicians as a flat EEG trace. This state also co-exists within the vicinity of the self-sustaining AI activity state, when $[O_2]_{Buffer}$ and $[K^+]_{Buffer}$ are near their normal values (Fig. 4a) and, as such, is a stable fixed point attractor. Moreover, this isoelectric state is bistable with the AI state, such that the observed dynamics depend on initial conditions. The isoelectric state is observed when the initial conditions are in a non-AI regime, otherwise the stable AI regime emerges.

Note that these conditions arise in the absence of an external stochastic drive. To assess the stability of the isoelectric state, we performed additional simulations in the setting where the network receives external stochastic drive. We find that the coexisting iso-electric state is only stable (non-spiking) in the presence of very small perturbations ($\lesssim 1.5\,\mu A/cm^2$ amplitude; Supplementary Fig. S3). External noise of greater amplitude shifts the dynamics to the AI regime (Supplementary Fig. S3b–d). In contrast, under normal physiological conditions in the AI state, the amplitudes of spontaneous post-synaptic currents are approximately 50–100 $\mu A/cm^2$ (Supplementary Fig. S4, left panels), hence up to 2 orders of magnitude stronger than required to disrupt the isoelectric state (Supplementary Fig. S4, right panels). Therefore, while the isoelectric state co-exists as an attractor in this region of parameter space, it has a very small basin of attraction and as such is unlikely to be observed under normal physiological conditions.

## Hypoxia triggers pathological dynamics
Next, we explored the departure of the model dynamics from the healthy regime as $[O_2]_{Buffer}$ and $[K^+]_{Buffer}$ deviate from their normal values.

**Pathological dynamics due to hypoxia.** Reducing $[O_2]_{Buffer}$ to simulate metabolic conditions of hypoxia in the model results in pathological dynamics. We quantified the network's synchronization using the Kuramoto order parameter ($\langle R(t)\rangle$; see Methods). The healthy regime exhibits low synchronization as expected for an AI state (Fig. 4b). Decreasing $[O_2]_{Buffer}$ results in a transition to highly synchronized dynamics, on the border between the AI and isoelectric states at higher $[K^+]_{Buffer}$ (Fig. 4b). We refer to this highly synchronized region as the pathological regime.

Inspection of the firing rate time series in the pathological regime reveals bursting dynamics (bursts interspersed with silence). However, the bursts are diverse in nature, forming two zones in the pathological regime (Fig. 4c). We show exemplar single-neuron membrane potential time series for all states in Supplementary Fig. S2. In the first zone, bursts take the form of neuronal avalanches in the firing rate time series (labeled BS state), similar to the pattern of BS seen in neonates recovering from an ischemic-hypoxic insult (e.g. birth asphyxia)[8,23]. In the second zone, bursts are composed of a series of synchronous spikes in the firing rate time series, similar to electrophysiological recordings of seizures (labeled SZ state). The transition between BS and SZ is not sudden but rather consists of an intermediate region where seizures and bursts occur simultaneously. In the present work, we do not explore this intermediate region and include it in the BS zone. Next, we examine the BS and SZ states separately.

**Hypoxia-induced BS.** In newborn infants, perinatal ischemia-hypoxia generates the pathological EEG phenomenon of BS[8]. In the model, we observe BS while reducing $[O_2]_{Buffer}$ for values of $[K^+]_{Buffer}$ higher than 17.5 mM (i.e., highly elevated $K^+$) (Fig. 4c). The BS regime exhibits

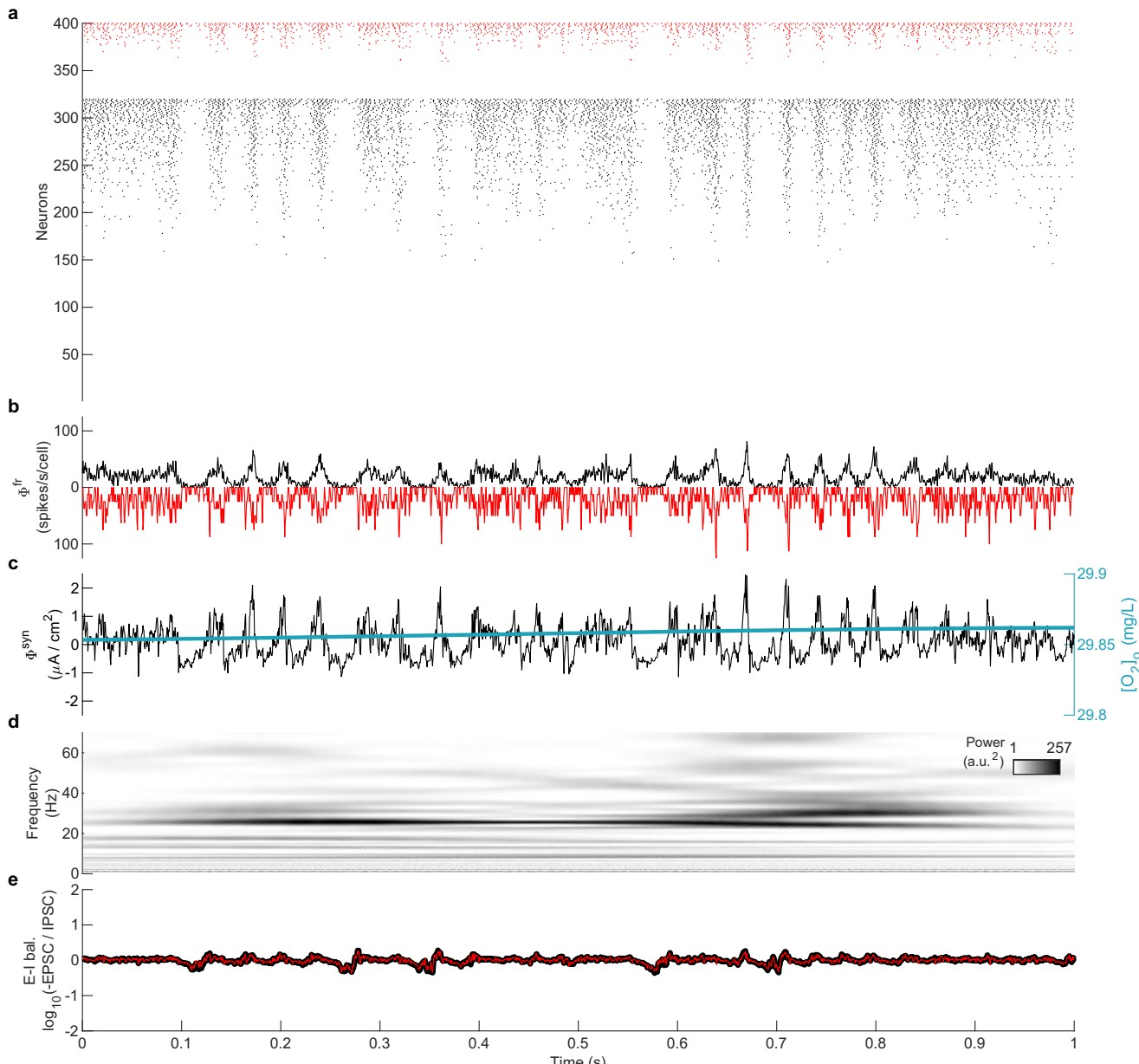

**Fig. 3 | Self-sustaining asynchronous irregular activity during plentiful oxygen supply.** **a** Raster plot showing asynchronous and irregular individual spiking dynamics ($CV_{ISI} = 1.01$, $CC = 0.04$) of 320 excitatory neurons (black) and 80 inhibitory neurons (red), sorted by firing rate. **b** Average of the instantaneous firing rates ($\Phi^{fr}(t)$) of excitatory (black) and inhibitory (red; plotted inverted) neurons. Each neuron's firing rate was calculated using a moving window of 1 ms such that there is 1 or 0 spike in a window. **c** Mean post-synaptic current $\Phi^{syn}(t)$ of excitatory neurons. Blue line shows the average local extracellular oxygen concentration $[O_2]_o$. **d** Spectrogram showing the wavelet-based time-frequency distribution of $\Phi^{fr}(t)$. **e** Excitation-inhibition (E-I) balance calculated as the logarithm (base 10) of the ratio of (-ve) excitatory post-synaptic current (EPSC) and inhibitory post-synaptic current (IPSC), shown for inputs to excitatory (black) and inhibitory (red) neurons. A value of 0 indicates balance. Small fluctuations away from zero are common to both populations.

bursts with a wide range of degrees of synchronization (Fig. 4b). A gradual increase in a narrow range of $[O_2]_{Buffer}$ results in phase-transitions from ordered BS to scale-free BS to disordered BS (Supplementary Fig. S6).

An example of modeled BS dynamics is shown in Fig. 5 at $[K^+]_{Buffer} = 20$ mM and $[O_2]_{Buffer} = 7.05$ mg/L. For each burst, neurons fire in a relatively synchronized ($CC = 0.18$) but irregular ($CV_{ISI} = 11.32$) fashion, before falling silent for a variable duration until the next burst (Fig. 5a). These synchronized and irregular firings give bursts their long duration and hence dynamic spectra are dominated by low frequencies, lacking harmonic structure (Fig. 5d). The number of neurons involved in each burst is highly variable, reflected in highly variable amplitudes of LFP/EEG proxy signals (Fig. 5b, c).

Due to the low availability of $O_2$ from $[O_2]_{Buffer}$, the local extracellular $[O_2]_o$ dynamics are tightly intertwined with the corresponding neural activity (Fig. 5c). While on the time scale of hundreds of seconds the model exhibits bursts and suppressions as is characteristic of burst suppression in EEG. When viewed in more detail, these bursts comprise clusters of briefer bursts. A single brief burst may not necessarily involve all neurons, and is typically followed by another brief burst involving different neurons after a short suppression phase, giving rise to a cluster of bursts. The clusters of bursts appear around local maxima of $[O_2]_o$ (Fig. 5c). $[O_2]_o$ decreases with each burst in the cluster. When more or less every neuron has participated in the cluster, a long suppression phase is observed during which oxygen replenishes (Fig. 5c).

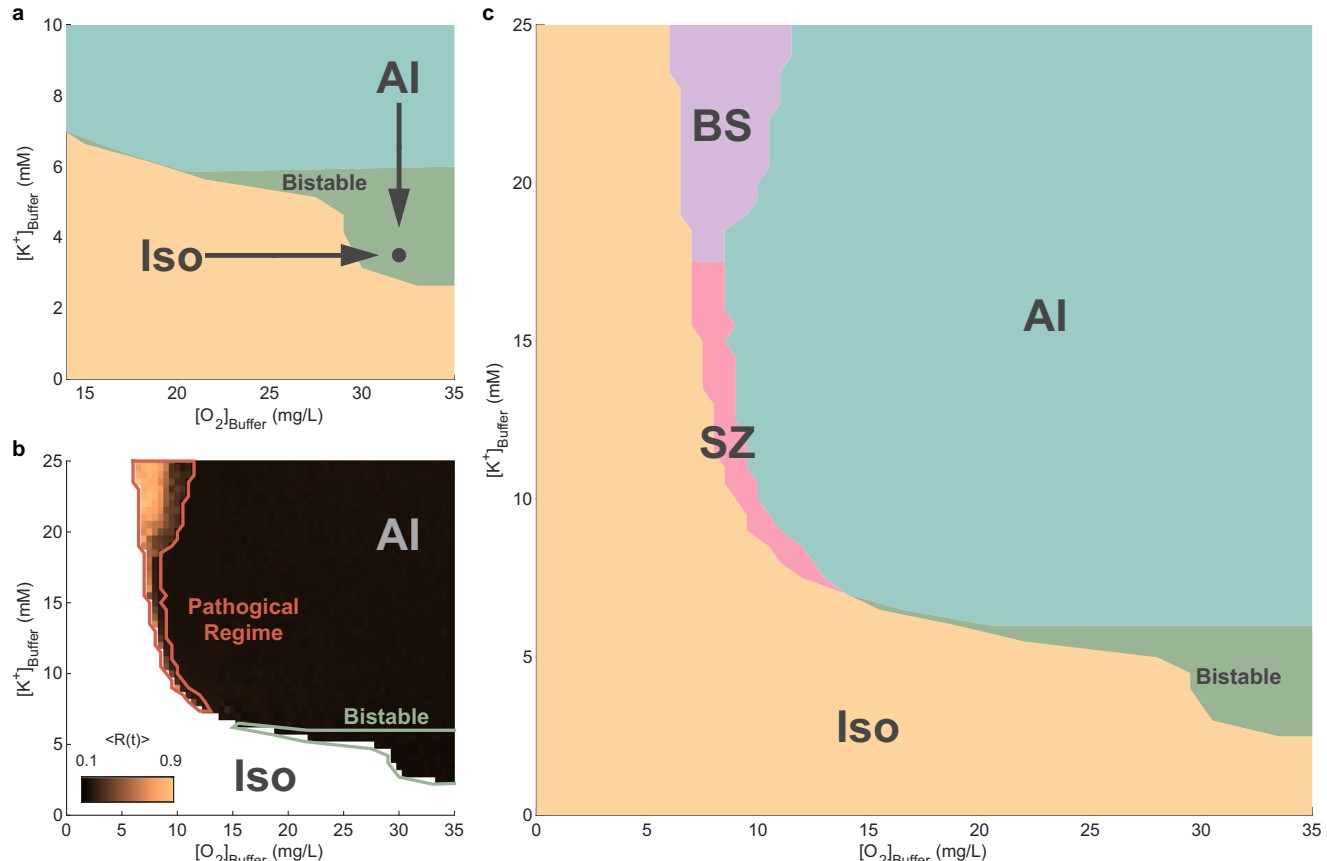

**Fig. 4 | Unification of four pathological states and healthy state in the [K⁺]$_{Buffer}$-[O₂]$_{Buffer}$ plane. a** At normal [O₂]$_{Buffer}$ and [K⁺]$_{Buffer}$ (black dot), the dynamics show the asynchronous irregular (labeled AI) state (Fig. 3) bistable with the isoelectric state (labeled Iso). **b** Kuramoto order parameter $\langle R(t) \rangle$ quantifying synchronization across the [O₂]$_{Buffer}$-[K⁺]$_{Buffer}$ plane. The high $\langle R(t) \rangle$ regime is labeled Pathological Regime. **c** The pathological regime comprises two pathological states: a seizure state (SZ, pink), and a hypoxic burst-suppression state (BS, violet).

These BS dynamics derive from the complex interplay of neuronal and network dynamics across three distinct time scales (Supplementary Fig. S5). The fast timescale of individual spikes ($V$) derives from the classic Hodgkin-Huxley-type membrane capacitance (Supplementary Fig. S5b). The second time scale corresponds to the repetitive firing of many cells within a burst timelocked to the recovery dynamics of potassium (Supplementary Fig. S5c). The third time scale reflects the interplay of slow metabolic ([O₂]$_o$, Supplementary Fig. S5d) and ionic ([Na⁺]$_i$, Supplementary Fig. S5e) processes, yielding the duration of the bursts and the interval between them. The second and third time scales emerge because the ionic concentrations change in response to neuronal firing. The characteristics of the network bursts (such as size and duration) depend on the number of recruited neurons and the timing of the onsets of the bursts in relationship to the recovery of [O₂]$_o$ and [Na⁺]$_i$. As the number of recruited neurons varies within each network burst, the system fluctuates between small (few neurons recruited) and large bursts (most neurons recruited). Small bursts typically occur when a burst is initiated when the metabolic states of most of the system's neurons are still recovering from the previous burst (see example in Supplementary Fig. S5 at ~150 s). Conversely, larger bursts occur when [O₂]$_o$ and [Na⁺]$_i$ have recovered in most neurons (see example in Supplementary Fig. S5 at ~100 s). The time scales associated with BS bursts are longer than those of the activity fluctuations in the AI state, which do not exhibit large network-wide changes in ionic concentrations and thus do not engage these slower time scales.

In a clear departure from the AI state (Fig. 3e), the E-I balance is highly disrupted in the BS state (Fig. 5e). The bursts exhibit large deflections in E-I balance, alternating between excitation-dominated and inhibition-dominated fluctuations.

**Hypoxia-induced seizure-like activity (SZ).** Decreasing [O₂]$_{Buffer}$ in combination with values of [K⁺]$_{Buffer}$ between ~7 and 17 mM (i.e., moderately elevated K⁺) results in the emergence of seizure-like activity (SZ regime, Fig. 4c). This activity exhibits several features present in human seizures. First, there is substantial activation of neurons across the network during the model seizures (Fig. 6a). The example SZ state shown at [K⁺]$_{Buffer}$ = 8 mM and [O₂]$_{Buffer}$ = 11.33 mg/L has $CV_{ISI}$ ~ 3.7 and CC ~ 0.15. This implies that the SZ state is more irregular but also more synchronous than the AI state. The increased synchrony translates into higher amplitudes for the LFP/EEG traces: the firing rate time series (Fig. 6b) and PSC time series (Fig. 6c) in the SZ state are substantially higher than in the AI state; large amplitude LFP/EEG is a hallmark of epileptic seizures. We also note that unlike the AI state, the seizure depletes locally available oxygen. Almost every neuron participates during a seizure event. A suppression phase occurs post seizure termination during which time oxygen replenishes (Fig. 6c).

A rapid slowing of frequencies (chirps) and their harmonics in the time-frequency spectrogram is a highly specific and sensitive signature of epileptic activity in adults[29]. Here we find that the model seizures exhibit this slowing of frequencies and their harmonic content (Fig. 6d). Similar to the BS state, E-I balance in the SZ state is highly disrupted (Fig. 6e). This observation is consistent with previous reports of E-I imbalance during epileptic seizures in humans[25].

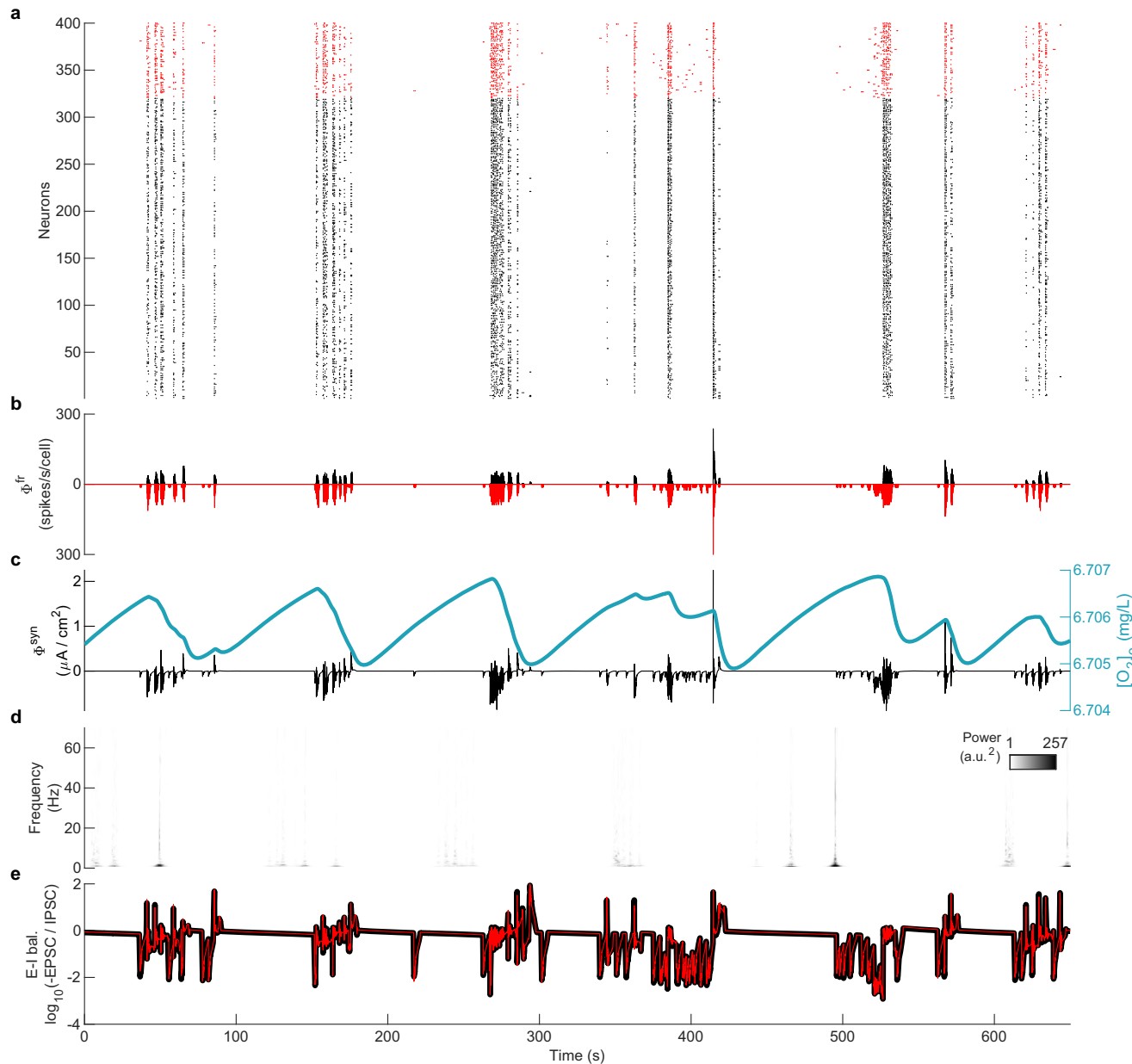

**Fig. 5 | Hypoxia-induced burst-suppression (BS) state. a** Raster plot showing bursts of activity in the BS state at $[K^+]_{Buffer} = 20$ mM and $[O_2]_{Buffer} = 7.05$ mg/L, sorted by firing rate. **b** Average of the instantaneous firing rates ($\Phi^{fr}(t)$) of excitatory (black) and inhibitory (red; plotted inverted) neurons. **c** Mean post-synaptic current $\Phi^{syn}(t)$ of excitatory neurons. The blue line shows the average local extracellular oxygen concentration $[O_2]_o$. **d** Spectrogram showing wavelet-based time-frequency distribution of $\Phi^{fr}(t)$. **e** Excitation-inhibition (E-I) (im)balance shown for inputs to excitatory (black) and inhibitory (red) neurons.

## Role of potassium in recovery from hypoxic insults

We next simulated the conditions for a successful recovery following a hypoxic insult—that is, as a return to the healthy AI state following imposition of brief hypoxia. A hypoxic insult was introduced by decreasing $[O_2]_{Buffer}$ from its normal value of 32 mg/L to values where the network enters the isoelectric state. Because extracellular $K^+$ is known to increase in the brain post hypoxic insults[30–42], we also explored the effect of increased $[K^+]_{Buffer}$ during the post hypoxic recovery.

**Conditions for recovery at normal $[K^+]_{Buffer}$.** In the simplest case where $[K^+]_{Buffer}$ stays at its normal value (Fig. 7a), healthy activity can persist for short hypoxic insults but not for insults of duration >10 s (Fig. 7b, c). Recovery from an insult of durations <10 s depends on the severity of the insult: The maximum survivable duration of insult

decreases with the magnitude of the decrease in $[O_2]_{Buffer}$ ($\Delta[O_2]_{Buffer}$) from its normal value (Fig. 7c). This reproduces the expected behavior that mild hypoxia can be tolerated for longer than severe hypoxia.

**Conditions for recovery via high $[K^+]_{Buffer}$.** Increasing extracellular $K^+$ has been shown to assist recovery in the heart cells of guinea-pigs deprived of oxygen[43]. We hence examined recovery from hypoxia via an increase of $K^+$. We observed that recovery to the AI state via high $[K^+]_{Buffer}$ (Fig. 7d–f) occurs if the duration of high $[K^+]_{Buffer}$ exceeds a minimum survivable duration (Fig. 7e, f). The minimum survivable duration decreases with the magnitude of the increase in $[K^+]_{Buffer}$ ($\Delta[K^+]_{Buffer}$) from its normal value (Fig. 7f). That is, in this scenario, inducing high $K^+$ is protective against a poor post-hypoxia outcome. Two other features of the dynamics concord with phenomena seen experimentally: brief periods of synchronous activity occur at the

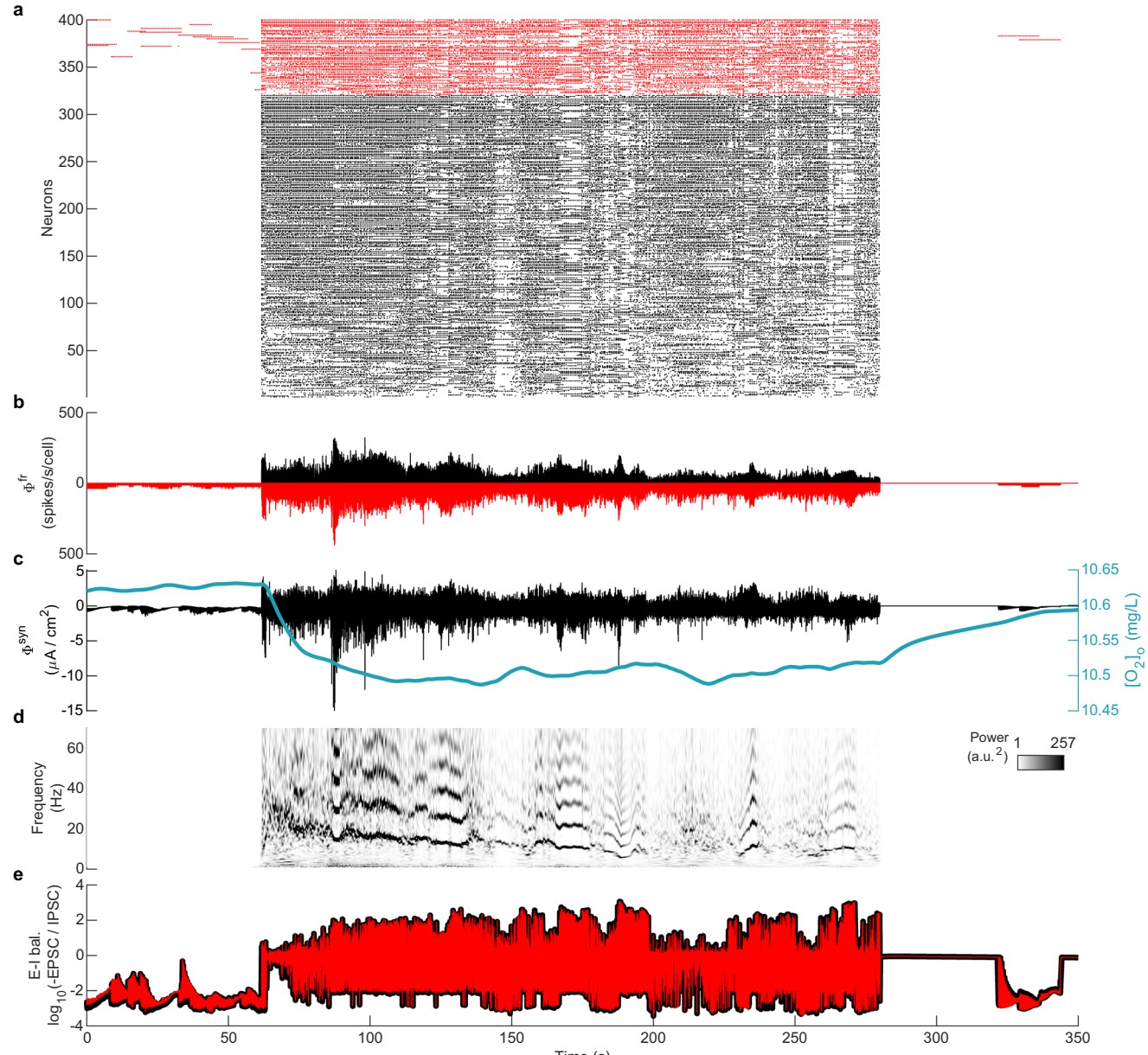

**Fig. 6 | Hypoxia-induced seizure (SZ) state. a** Raster plot showing high firing rate activity in the SZ state at $[K^+]_{Buffer} = 8$ mM and $[O_2]_{Buffer} = 11.33$ mg/L, sorted by firing rate. **b** Average of the instantaneous firing rates ($\Phi^{fr}(t)$). **c** Mean post-synaptic current $\Phi^{syn}(t)$ of excitatory neurons. Blue line shows the average local extracellular oxygen concentration $[O_2]_o$. **d** Spectrogram showing wavelet-based time-frequency distribution of $\Phi^{fr}(t)$. **e** Excitation-inhibition (E-I) (im)balance shown for inputs to excitatory (black) and inhibitory (red) neurons.

transition to isoelectric activity and during the recovery of normal irregular activity upon re-oxygenation: These are similar to the wave of death and wave of resuscitation that have been observed in animal neurophysiological recordings following hypoxia[44–46].

**Recovery from hypoxia via BS at high $[K^+]_{Buffer}$.** Neonates recovering from ischemia-hypoxia exhibit periods of BS often lasting many hours[8,47]. In our model the BS pathological regime occurs at high $[K^+]_{Buffer}$ (Fig. 4c) supporting the view that an increase in $K^+$ following hypoxia may play a protective role. Therefore, we simulated hypoxic recovery via the BS regime (Fig. 7g). This was achieved by moving the $[K^+]_{Buffer}$ and $[O_2]_{Buffer}$ parameters into the BS regime following a brief hypoxia (Fig. 7g). We then gradually increased $[O_2]_{Buffer}$ to simulate reperfusion of the neural tissue (Fig. 7g, bottom panel). We select a range of $[O_2]_{Buffer}$ where the observed

dynamics span from ordered BS to scale-free BS and disordered BS (Fig. 4b and Supplementary Fig. S6).

After a long quiescent period, the model exhibits bursts interspersed with silent periods (Fig. 7h). As $[O_2]_{Buffer}$ gradually increases, the inter-burst-interval decreases, hence increasing the density of bursts (Fig. 7h). This is the characteristic feature of recovery from ischemic-hypoxic insults widely observed in infant EEG[8].

**Mechanisms for the role of $K^+$ in successful recovery.** In sum, we observe an apparently protective role of increased $K^+$ (whether or not BS is involved). To gain an understanding of this, we performed numerical simulations (Fig. 7i–k) straddling the maximum survivable duration of hypoxia, which is in between the hypoxia duration of 1 s (Fig. 7b, top) and 2 s (Fig. 7b, bottom). During the period of hypoxia, $[O_2]_o$ decreases, and $[K^+]_o$ and $[Na^+]_i$ increase due to the impact of low

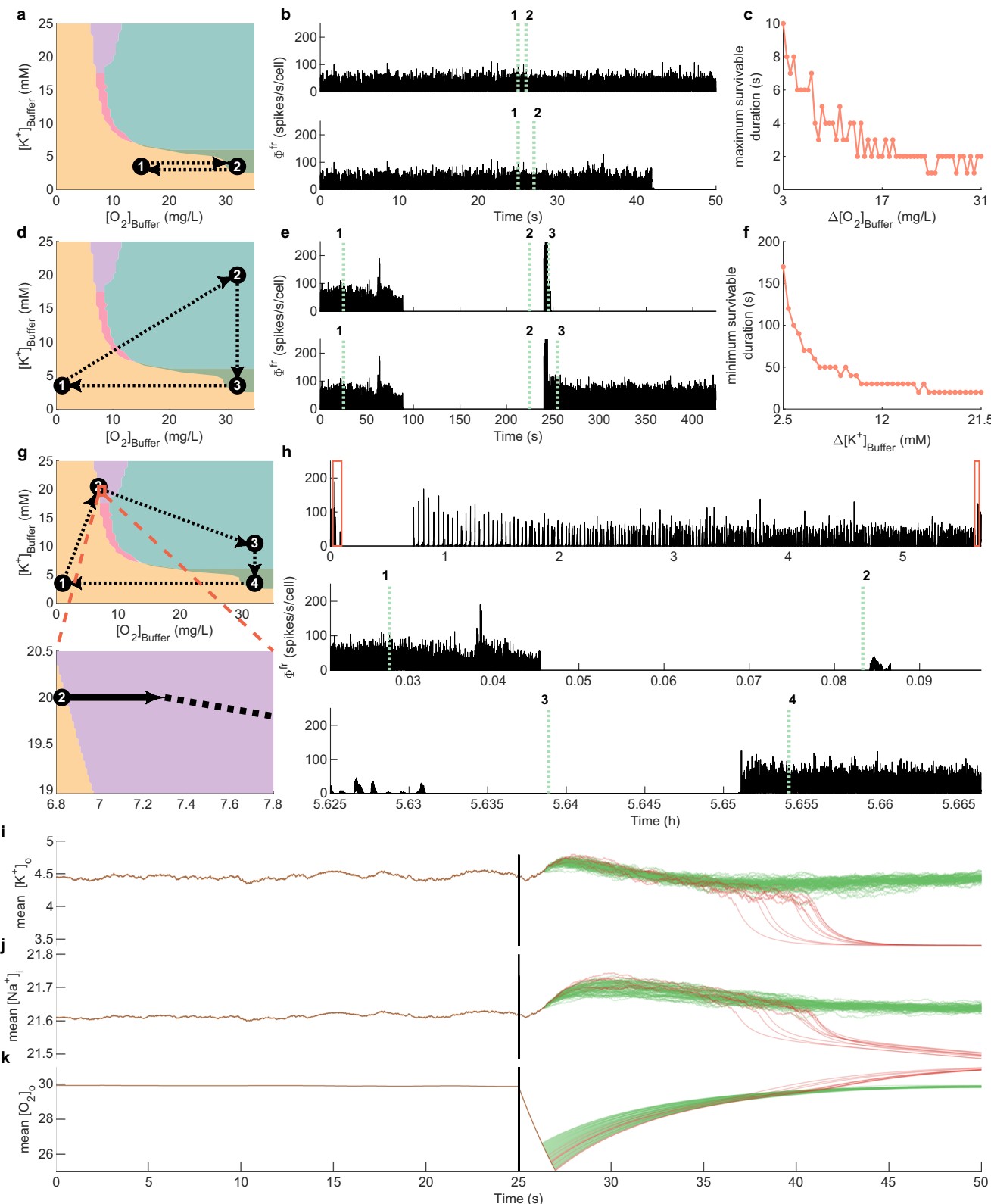

extracellular oxygen on the Na⁺-K⁺ pumps. When the hypoxia ends, $[O_2]_o$ slowly recovers allowing Na⁺-K⁺ pumps to restore $[Na^+]_i$ and $[K^+]_o$. Irrespective of whether activity persists or ceases, $[K^+]_o$ returns to its pre-hypoxia values prior to the return of $[Na^+]_i$ because of the differences in their respective time scale parameters. Thereafter, $[K^+]_o$ continues to decrease below its pre-hypoxia range. This is because as the Na⁺-K⁺ pump continues to restore $[Na^+]_i$, it exchanges 3 Na⁺ for 2 K⁺, decreasing $[K^+]_o$ below its equilibrium value. This decreases the

reversal potential of K⁺ ($E_K$), and, therefore also decreases the resting membrane potential. Numerical simulations (Fig. 7i–k) suggest that if the over-correction of K⁺ is too large, neural activity in the system exhibits the delayed "collapse"−that is, it suddenly converges to the isoelectric state. Conversely, if the hypoxia is sufficiently brief so that sodium recovers before any overcorrection of K⁺, the system remains in the AI state. The critical value of K⁺−the separatrix−appears to be ~4 mM.

**Fig. 7 | Trajectories following hypoxic insults. a**–**c** Recovery failure. **a** Rapid $[O_2]_{Buffer}$ decrease from the healthy state to a hypoxic isoelectric state (1), followed by delayed rapid return (2). **b** Network firing rate for brief (1 s, top) and long (2 s, bottom) hypoxic insults. Dotted lines indicate start (1) and end (2) of hypoxia. **c** Maximum duration at low $[O_2]_{Buffer}$ (labeled as maximum survivable duration) for which dynamics return to the AI state as a function of hypoxic insult depth $\Delta[O_2]_{Buffer}$. **d**–**f** Recovery via high $[K^+]_{Buffer}$ **d** Rapid $[O_2]_{Buffer}$ decrease from the healthy state to a severe hypoxic isoelectric state (1), followed by a high-$[K^+]_{Buffer}$ AI state (2), then recovery (3). **e** Network firing rate for brief (20 s, top) and long (30 s, bottom) elevated $[K^+]_{Buffer}$ (=19.5 mM) periods. Dotted lines indicate the start (1) and end (2) of hypoxia, and start (2) and end (3) of high $[K^+]_{Buffer}$, returning to the healthy state (3). **f** Minimum duration of high $[K^+]_{Buffer}$ to return to the AI state (labeled as minimum survivable duration) as a function of increase in $[K^+]_{Buffer}$ from its normal value 3.5 mM ($\Delta[K^+]_{Buffer}$). **g**, **h** Recovery via BS at high $[K^+]_{Buffer}$. **g** Top:

Rapid $[O_2]_{Buffer}$ decrease from the healthy state to a severe hypoxic isoelectric state (1), followed by traversing the BS state (2), an intermediate $[K^+]_{Buffer}$ state (3), and recovery (4). Bottom: Zoom of the BS regime where $[O_2]_{Buffer}$ increases gradually over ~5 h (solid black arrow). **h** Network firing rate. Top: Entire trajectory. Red boxes denote zooms in panels below. Middle: Hypoxic insult (1) and start of the $[O_2]_{Buffer}$ increase during BS (2). Bottom: Final recovery. Dotted lines denote the rapid transition from the end of BS to the high $[K^+]_{Buffer}$ AI state (3) and rapid return to the healthy state (4). **i**–**k** Activity termination in panel (**b**) due to $K^+$ over-correction. Simulations for hypoxia durations between 1 s (cf. panel (**b**), top row) and 2 s (cf. panel (**b**), bottom row) hypoxia; onset denoted by black line. Activity survival shown in green, cessation in red. **i** Mean $K^+$ across neurons. **j** Mean $Na^+$ across neurons. **k** Mean $[O_2]_o$ across neurons. Shading in panels (**a**), (**d**), and (**g**) is as per Fig. 4.

## Inferred trajectories from neonatal BS data

Finally, we inferred trajectories in model parameter space from clinical EEG recordings of BS. For this, we restricted attention to the region in the parameter space with a diversity of BS patterns (Fig. 7g [bottom panel] and Supplementary Fig. S6). Each location in this region is represented as a vector of six burst metrics (Fig. 2i–t). Similarly, each 300 s non-overlapping window in an EEG epoch (Fig. 2a–h) is represented by a vector of six burst metrics. Trajectories were inferred by projecting the time series of empirical feature vectors onto the parameter space such that the difference between the modeled and empirical feature vectors is minimized subject to a constraint ensuring smoothness of the resulting trajectory (see Methods for details).

These parameter space trajectories were assessed in 17 infants who had developmental outcomes available: 10 infants with good recovery were either normal or had mild neuromuscular disorders (mild injury) by age 1–3 years, while 7 infants with poor recovery either died, had severe neuromuscular disorders (severe injury), or had a thalamic lesion. The parameters of an exemplar infant with a good recovery (Fig. 8a; left panel) yield a trajectory that moves incrementally (from blue to red) towards the healthy regime. The inferred parameters of an exemplar infant with poor recovery (Fig. 8b; left panel) yield a trajectory that dwells around the pathological values of $[K^+]_{Buffer}$ and $[O_2]_{Buffer}$. We also estimated average (median) trajectories (Fig. 8a, b; right panels) from the inferred trajectories of infants with good versus poor recovery after scaling individual trajectories to unit time. Overall we found that the trajectories for babies with good recovery tended to travel toward normal values of $[K^+]_{Buffer}$ and $[O_2]_{Buffer}$—i.e., toward the healthy region—whereas for babies with poor recovery the trajectories dwelled near the pathological region. Quantitatively, the mean $\Delta[K^+]_{Buffer}$ of the last epoch of babies with good outcome was significantly lower than that of babies with poor outcome (two-tailed *t*-test $p = 0.0077$, *t*-statistic $= -3.0756$, df $= 14.9771$, Fig. 8c). By repeating these analyses using the instantaneous power time series of $\Phi^{syn}$ (i.e., the square of the absolute values of the Hilbert-transform-derived analytical signal), we obtain trajectories that broadly resemble those derived from the $\Phi^{fr}$ time series. However, burst metrics are sensitive to the choice of the LFP proxy, resulting in some differences between the two sets of trajectories (Supplementary Figs. S7 and S8). Nevertheless, the inferred changes in potassium levels differentiating good versus poor outcomes remain largely preserved (Supplementary Fig. S9).

In addition, we examined potential redundancies between burst metrics by removing one parameter at a time and reconstructing the inferred trajectories using the remaining five metrics. We found that these five-parameter trajectories are broadly similar to their original six-parameter trajectories, implying partial redundancy, though there is no universally redundant parameter (Supplementary Figs. S10 and S11).

As a sanity check, we simulated time series using the inferred model parameter trajectories. For each window in the epochs of the

exemplar infant (Fig. 2), we generated time series of 5 min (matching the duration of the window) parameterized by the corresponding inferred model parameters (Fig. 8b; left panel). The model generated instantaneous power time series (Fig. 9a) that are in close agreement with the original EEG power (Fig. 9b). The sample windows in the epochs of exemplar data time series (Fig. 2 and 9a; green windows), and the same windows in the corresponding model-generated time series (Fig. 9b; pink windows) are also in close agreement in terms of qualitative closeness of time series and of the six-burst metrics (Fig. 9c–f). In the supplementary movies (Supplementary Movies 1–17), we show the evolution of trajectories, matching of simulated time series and data time series, and the matching of their corresponding six features. On the whole, the model and data agree well across the full set of time windows, although there is some variability. For example, trajectories for some of the infants with good recovery do not track towards the healthy region, and a trajectory for one infant with poor recovery appears to move towards healthy region. Nevertheless, the average trajectories follow the expected trend (Fig. 8a, b).

To assess the validity of the trajectory inference Algorithm 1, we inferred metabolic parameters from three distinct synthetic trajectories: a straight line, a kinked trajectory, and a loop (Supplementary Fig. S12a). We then estimated burst statistics from the simulated time series and used the same algorithm employed for the empirical data to infer the optimal parameter trajectory. We found that the inferred parameters captured the basic trends present in the ground truth synthetic trajectories and are distinguishable from one another (Supplementary Fig. S12b). Notably, the straight and kinked trajectories can be easily disambiguated from the looped curve, an important property observed in the good (straighter) versus poor (more looped) outcome neonates.

These proof-of-principle results support the possibility of model-based state estimation for tracking the evolution of brain dynamics following birth asphyxia.

## Discussion

Computational modeling of brain activity and function has overwhelmingly focused on complex neuronal activity considered in a metabolic vacuum—that is, without incorporating the close interdependence of neuronal activity and its metabolic support[23]. Both as a functional constraint—such as the energy sparing notion of sparse spiking[48]—and as a dynamic and coupled compartment—as explored here—biophysical modeling of neural systems needs to overcome this limitation. Here, using the resource-depleted state of perinatal hypoxia, we show that such an approach is able to capture key forms of normal and pathological activity, and identify key protective responses (such as increased $K^+$). As a proof-of-principle, we have shown that tracking the parameters of recovering versus poor-outcome neonates is possible, and yields predicted dynamics that indeed mimic those seen in the clinic. This argues for integrated models of neural-metabolic activity and suggests translational opportunities.

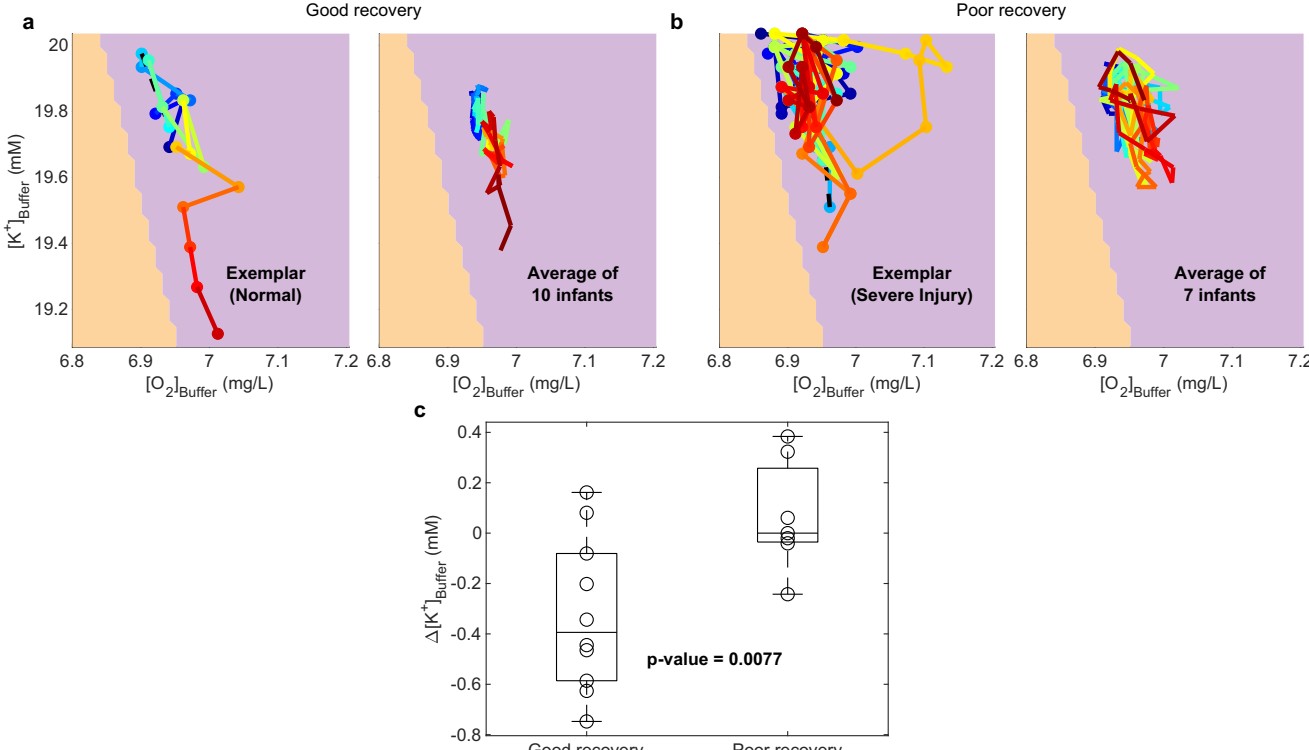

**Fig. 8 | Inferred trajectories in the model parameter space from neonatal data.**
**a** Neonates with good recovery outcome. **b** Neonates with poor recovery outcome. Colors denote time along the complete trajectories, from blue to red, with black dashed lines connecting epochs. Left panels show trajectories from an exemplar baby in each group. Right panels show the median trajectory for each group. **c** Box and whisker plots for $\Delta[K^+]_{Buffer}$ from the last epoch for infants with good ($n = 10$) versus poor recovery outcomes ($n = 7$). The $p$-value is for a two-sided $t$-test with unequal variances. Circles, all individual (per infant) $\Delta[K^+]_{Buffer}$ values; center line, median; box limits, upper and lower quartiles; whiskers, minimum and maximum. Shading in panels (**a**), (**b**) is as per Fig. 4.

To our knowledge, the pathological pattern of scale-free burst suppression due to ischemic-hypoxic insult seen in neonates[8,23] has not been previously modeled. We find an important role for coupling of metabolic variables with neuronal dynamics, as has been considered in several studies modeling anesthesia-induced burst suppression[9,10,49], and hypoxia-induced seizures[16,17]. We find that interplay between the local oxygen availability ($[O_2]_o$) and brain activity gives rise to three pathological regimes and a self-sustaining asynchronous irregular (AI) regime, depending on the values of $[K^+]_{Buffer}$ and $[O_2]_{Buffer}$.

Our exploration of recovery trajectories revealed that in addition to timely reoxygenation, an increase in $[K^+]_{Buffer}$ facilitates the restoration of healthy dynamics by preventing the over-correction of $[K^+]_o$ during re-oxygenation. This suggests that a substantial increase in potassium following hypoxia (as observed empirically[30–41,50]) could be a protective mechanism that brings the dynamics closer to the BS regime. In the BS regime, the estimated parameter trajectories inferred from EEG data suggest effective potassium clearance during reoxygenation for babies with good recovery, and its failure for babies with poor recovery. Various proteins facilitating potassium clearance are up-regulated after ischemic injury in astrocytes[51] in response to massive $[K^+]_o$ increase. Our modeling suggests an important role of these $K^+$-clearance mechanisms for infants with good recovery. However, these mechanisms apparently fail for babies with poor recovery. While mechanistic studies are clearly needed, this finding highlights the potential therapeutic insights provided by coupled neuronal-metabolic modeling.

Several limitations of the present work could be addressed through future work. First, several model assumptions could be revisited. Although $Na^+$-$K^+$ pumps account for a majority of the energy expenditure for signaling, it has recently been found that ~44% of the brain's energy is used to maintain the integrity of the synaptic vesicles

independent of signaling[52]. Therefore, further extensions could incorporate this non-signaling energy budget. Second, due to model complexity and computational constraints we only explored a network of 400 neurons. Scaling up to the large scales of EEG and whole-brain dynamics would likely be better suited to mean field models that pool the collective activity of neurons and hence reduce model dimensionality[49,53]. Such developments will be crucial to study spatial patterns[54] or the regional effects of local ischemic-hypoxic insults. Moreover, comparison with EEG (and other modalities) would be improved using a detailed forward (observation) model that maps neuronal variables to measured quantities (e.g., taking into account electrode geometry and tissue properties). Third, state estimation techniques (e.g., maximum a posteriori estimation, Markov chain Monte Carlo, variational inference, etc.) could be used to find better estimates of parameters than the current look-up method, and could also return the full parameter probability likelihoods. Such methods have recently been applied in developing computational modeling of epilepsy into a clinically useful tool[55,56]. Fourth, changes in pH and temperature are important clinical factors during recovery[57,58]. Future modeling steps could include these variables and their effects on dynamics; for instance, hypothermic cooling is part of the standard care following perinatal asphyxia. Future work could also use machine learning tools to identify optimal personalized or global sets of burst metrics. Finally, we indirectly inferred oxygen and ionic dynamics, which can be measured directly in experimental systems, enabling more powerful calibration and validation of the model. In humans, further calibration would be possible for a large scale version of the model using positron emission tomography to infer metabolic activity directly, in concert with observed in vivo neural activity.

Seizures and BS both pose significant metabolic challenges to the cortex and can occur together after birth hypoxia. Our model

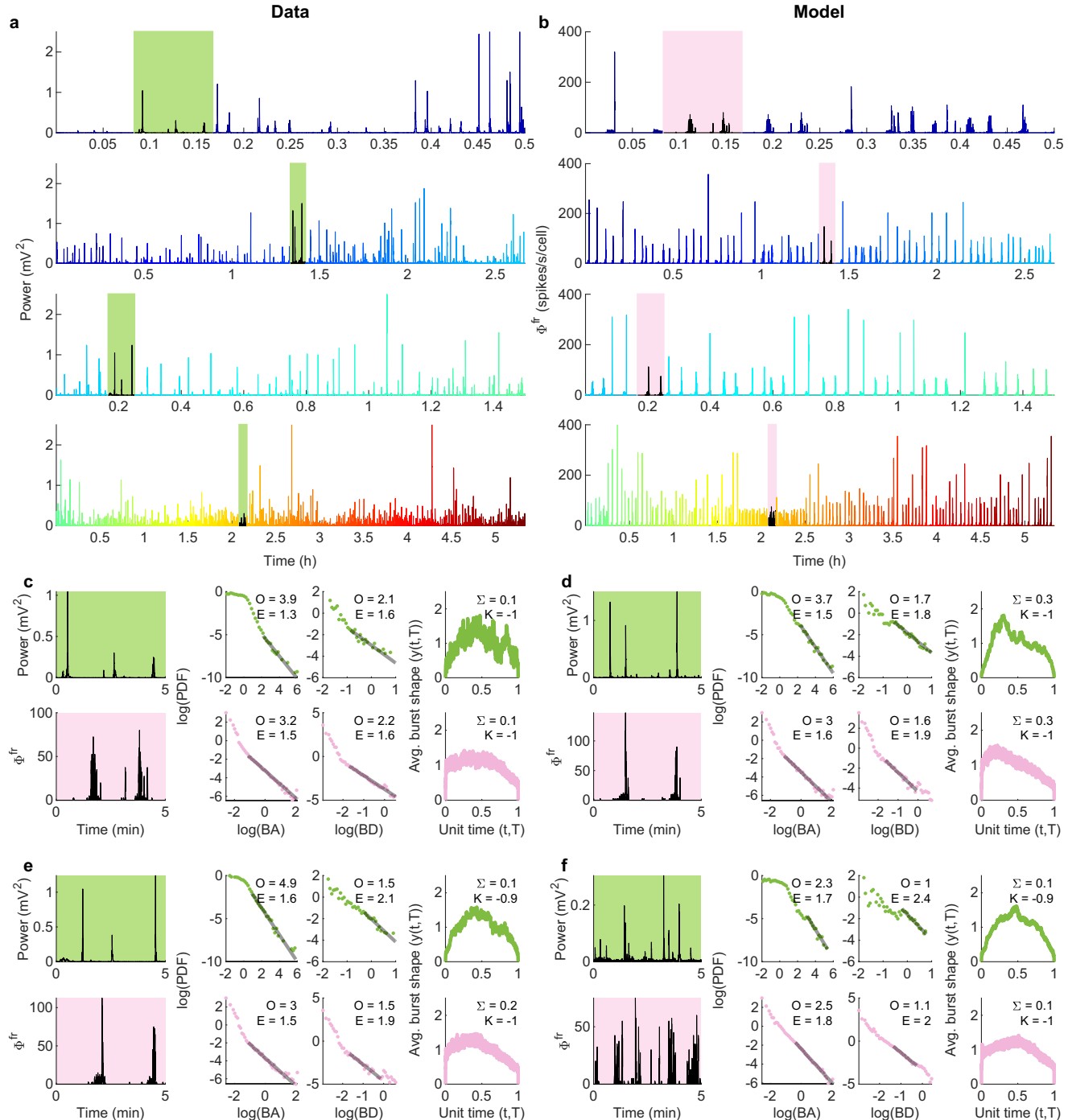

**Fig. 9 | Model generated time series using inferred parameters from data.**
**a** Time series of an infant with poor recovery outcome (shown in Fig. 2). **b** Model simulated time series using the inferred parameters in Fig. 8b left panel. Green and pink boxes in a and b indicate 5 min windows analyzed in (**c**–**f**). Color in the time series becomes warmer with time as per the accompanying trajectory in Fig. 8b left panel. **c**–**f** Burst metrics of the four sample windows highlighted in panels (**a**) and (**b**), expanded here in the left panels (data in green, model in pink). Right panels show probability density functions (PDFs) for burst areas (BA) and durations (BD), along with average burst shapes.

can quantitatively reproduce essential seizure properties, including high-amplitude waveforms, variable durations, oscillations, and slowing of frequency harmonics. For quantitative model parameter estimation, it remains an open question whether six BS features used in this study capture clinically meaningful properties of seizures. While it is likely that the six features capture some fundamental characteristics such as duration, we expect that the oscillatory nature of seizures contains unique information that would be best captured by measures sensitive to the fundamental

frequency, harmonics, and gradual intra-ictal slowing (chirps), which will be explored in future work.

Validation of model predictions and calibration of model parameters can be achieved through future experiments. Although we are not aware of experiments that have investigated changes in neural activity across multiple controlled values of $[K^+]_{Buffer}$ and $[O_2]_{Buffer}$, independent evidence supports the emergence of seizure-like activity for higher $[K^+]_{Buffer}$[59,60], and conversely, for lower $[O_2]_{Buffer}$[22]. Moreover, substantial evidence suggests that hypoxia increases extracellular

potassium[9,30–42,61–66], thus supporting our fundamental approach that understanding the response to hypoxia activity requires a joint modeling of both $[K^+]_{Buffer}$ and $[O_2]_{Buffer}$. More broadly, we suggest that systematically mapping out the $[K^+]_{Buffer}$-$[O_2]_{Buffer}$ plane in experimental systems could prove fruitful in understanding dynamics under metabolically challenged conditions and provide independent validation of our model.

We have exploited the dramatic metabolic depletion that occurs following birth asphyxia to understand models of coupled neuronal-metabolic activity. However, coupling between metabolic and neuronal dynamics is a growing area applicable to many brain states, including other pathological conditions such as adult stroke, and the altered neurovascular coupling in dementia. The brain's energy budget also imposes a strong constraint on the healthy brain[12,52,67,68], suggesting that future extensions could extend beyond pathological conditions to more deeply understand how energy constraints shape healthy neural activity.

## Methods

### Computational model

Our model comprises a network of excitatory and inhibitory Hodgkin-Huxley (HH) type neurons coupled with local $O_2$ dynamics via sodium-potassium (Na+-K+) pumps. It has previously been shown that a single neuron with these dynamics generates bursts when the supply of $O_2$ is constricted[16,17]. These bursts are similar to the observed seizures induced by hypoxia in rat hippocampal slices[16,22]. We use this model as the starting point of our exploration.

The Wei et al.[16] model was built on an earlier model incorporating sodium (Na+) and potassium (K+) concentration dynamics into the Hodgkin-Huxley equations[21]. Wei et al.[16] bidirectionally coupled metabolic resources ($O_2$) to the membrane potential ($V$) via electroneutral[69] effects of Na+-K+ pumps, maintaining the resting-state ion concentrations (Fig. 1i). The electroneutral effects are the indirect effects of Na+-K+ pumps of a neuron and its neighboring glia on the membrane potential of the neuron by regulating ion concentrations (Fig. 1i). Electrogenic effects of the pump directly influence the membrane potential due to an outward sodium current that make the membrane potential more negative, as Na+-K+ pumps transfer 3 Na+ ions outside the cell for every 2 K+ ions inside the cell. These electrogenic effects are considered in the later versions of the model[17]. In this paper, we consider the electroneutral effects of the Na+-K+ pumps. Na+-K+ pumps maintain homeostasis of the resting membrane potential of neurons and glia by replenishing the intracellular K+ and extracellular Na+ discharged during action potentials and synaptic transmission. This process entails moving ions against their concentration gradient, and requires a consistent supply of energy (ATP/$O_2$) (Fig. 1i). We model Na+-K+ pump currents of neuronal ($I_{pump}$) and glia cells ($I_{gliapump}$) as sigmoidal functions of intracellular Na+ concentration ($[Na^+]_i$) and extracellular K+ concentration ($[K^+]_o$), such that

$$I_{pump} = \frac{\rho}{\left(1+ \exp\left\{(25 - [Na^+]_i)/3\right\}\right)\left(1+ \exp\{5.5 - [K^+]_o\}\right)}, \quad (1)$$

$$I_{gliapump} = \frac{\rho}{3\left(1+ \exp\left\{(25 - [Na^+]_{gi})/3\right\}\right)\left(1+ \exp\left\{5.5 - [K^+]_o\right\}\right)}, \quad (2)$$

where $[Na^+]_{gi}$ is the intracellular Na+ concentration of glia cells, and $\rho$ is the maximum value of the sigmoid. Here $\rho$ is itself modeled as a sigmoidal function of the extracellular $O_2$ concentration ($[O_2]_o$), obeying

$$\rho = \frac{\rho_{max}}{1+ \exp\{(20 - [O_2]_o)/3\}}, \quad (3)$$

where $\rho_{max}$ is the maximum pump rate attained in the fully oxygenated state. These sigmoidal relationships for $I_{pump}$ and $I_{gliapump}$ increase the

pump currents when $[Na^+]_i$ (or $[Na^+]_{gi}$) or $[K^+]_o$ increase due to action potential generation and synaptic transmission, thus increasing the energy demand. The role of oxygen is to limit the maximum pump current when the locally available $[O_2]_o$ is low (i.e., hypoxia), thus limiting the energy demand.

The dynamics of $[O_2]_o$ depend on $O_2$ consumption by $I_{pump}$ and $I_{gliapump}$, such that as $I_{pump}$ and $I_{gliapump}$ increase, $[O_2]_o$ decreases. Thus, $I_{pump}$ and $I_{gliapump}$ represent energy demand. The supply of $O_2$ locally to a neuron is modeled as diffusion from the cerebral circulation, assumed to be a metabolic reserve with fixed concentration $[O_2]_{Buffer}$[16]. Incorporating this, the $[O_2]_o$ dynamics around a single neuron obeys.

$$\frac{d[O_2]_o}{dt} = -\alpha\lambda(I_{pump} + I_{gliapump}) + \epsilon_o([O_2]_{Buffer} - [O_2]_o), \quad (4)$$

where $\alpha$ is the conversion factor between pump current (mM/s) and oxygen consumption rate (mg/L/s) (see ref. 16 for details), $\lambda$ is the relative cell density between excitatory and inhibitory neurons, and $\epsilon_o$ is the oxygen diffusion rate.

The ion concentrations depend on the ionic currents in and out of the cell, with $[K^+]_o$ and $[Na^+]_i$ obeying

$$\frac{d[K^+]_o}{dt} = \gamma\beta I_K - 2\beta I_{pump} - I_{glia} - 2I_{gliapump} - \epsilon_k([K^+]_o - [K^+]_{Buffer}), \quad (5)$$

$$\frac{d[Na^+]_i}{dt} = -\gamma I_{Na} - 3I_{pump}, \quad (6)$$

where $\gamma$ is the unit conversion factor[16,21], $\epsilon_k$ is the potassium diffusion rate[21], and $\beta$ is the ratio of the intracellular volume to the extracellular volume. $I_{glia}$ is the current due to glial uptake of the surrounding potassium governed by

$$I_{glia} = \frac{G_{glia}}{1.0 + \exp\{(18 - [K^+]_o)/2.5\}}, \quad (7)$$

where $G_{glia}$ is the glial uptake strength of potassium.

For potassium, we also model diffusion from blood vessels and surrounding tissues into the extracellular space. These distal sources have previously been modeled as a potassium reserve with fixed concentration $[K^+]_{Buffer}$[16,21]. The parameter $[K^+]_{Buffer}$ is an effective value describing the collective buffering capacity of various sources of potassium, particularly crucial during metabolically challenged states. For example, during hypoxic insult, a decrease in potassium concentration of the tissue and its simultaneous increase in surrounding areas have been observed in vitro[30–34] and in vivo[35–37]. Hypoxia also induces a five-to-ten-fold increase in potassium concentration more distally in the subarachnoid fluid[38,39], with moderate increases in the blood plasma[38,40], cisterna magna fluid[38,40], cortical cerebrospinal fluid[41], and on the cortical surface[42]. It has also been observed that ATP-sensitive potassium channels ($K_{ATP}$) open during metabolically challenged conditions leading to the leakage of potassium outside the cell[9,61–63]. Along with $K_{ATP}$ channels, K+ also leaks from calcium-sensitive potassium channels which activate with the increase in the entry of calcium to the cell during hypoxia[62,64–66]. All these sources, along with the vasculature, contribute to the buffering capacity parameterized by $[K^+]_{Buffer}$ in our model.

All else being equal, increasing $[K^+]_{Buffer}$ increases $[K^+]_o$ (Eq. (5)), which increases the neural and glial Na+-K+ pump currents (Eqs. (1) and (2)). This in turn increases $O_2$ consumption (Eq. (4)); i.e., increases in $[K^+]_{Buffer}$ increase the baseline energy demand on top of which are superimposed the dynamically changing demands, which derive from neural firing.

Assuming that the inward flow of $Na^+$ is compensated by the outward flow of $K^{+}$[16,21], the intracellular concentration of $K^+$ ($[K^+]_i$) obeys

$$[K^+]_i = 140 + (18 - [Na^+]_i), \tag{8}$$

and assuming the total amount of $Na^+$ is conserved[16,21], the extracellular concentration of $Na^+$ ($[Na^+]_o$) obeys

$$[Na^+]_o = 144 - \beta([Na^+]_i - 18). \tag{9}$$

The extracellular and intracellular concentrations of $Na^+$, $K^+$, and $Cl^-$ determine the reversal potentials of the respective ions ($E_{Na}$, $E_K$, and $E_{Cl}$) governed by the following Nernst equations:

$$E_{Na} = 26.64 \ln \frac{[Na^+]_o}{[Na^+]_i}, \tag{10}$$

$$E_K = 26.64 \ln \frac{[K^+]_o}{[K^+]_i}, \tag{11}$$

$$E_{Cl} = 26.64 \ln \frac{[Cl^-]_i}{[Cl^-]_o}. \tag{12}$$

The reversal potentials shape the respective ion currents passing through the voltage-gated ion channels, such that

$$I_{Na} = G_{Na} m^3 h (V - E_{Na}) + G_{NaL}(V - E_{Na}), \tag{13}$$

$$I_K = G_K n^4 (V - E_K) + G_{KL}(V - E_K), \tag{14}$$

$$I_{Cl} = G_{ClL}(V - E_{Cl}), \tag{15}$$

where $G_{Na}$, $G_{NaL}$, $G_K$, and $G_{KL}$ represent conductances of sodium and potassium currents and their respective leak currents, $G_{ClL}$ is the conductance of the leak chloride current, $V$ is the membrane potential, $m$ and $n$ are activation gating variables for sodium and potassium channels, and $h$ is the inactivation gating variable for sodium channels. The dynamics of the gating variables are governed by

$$\frac{dm}{dt} = \alpha_m(1 - m) - \beta_m m, \tag{16}$$

$$\frac{dh}{dt} = \alpha_h(1 - h) - \beta_h h, \tag{17}$$

$$\frac{dn}{dt} = \alpha_n(1 - n) - \beta_n n, \tag{18}$$

where parameters $\alpha_m$, $\alpha_n$, $\alpha_h$, $\beta_m$, $\beta_n$, and $\beta_h$ are the opening and the closing rates of the ion channel state transitions. These rates depend on membrane potential ($V$ in mV) according to

$$\alpha_m = 0.32 \frac{V + 54}{1 - \exp\{-(V + 54)/4\}}, \tag{19}$$

$$\beta_m = 0.28 \frac{V + 27}{\exp\{(V + 27)/5\} - 1}, \tag{20}$$

$$\alpha_n = 0.032 \frac{V + 52}{1 - \exp\{-(V + 52)/5\}}, \tag{21}$$

$$\beta_n = 0.5 \exp\{-(V + 57)/40\}, \tag{22}$$

$$\alpha_h = 0.128 \exp\{-(V + 50)/18\}, \tag{23}$$

$$\beta_h = \frac{4}{1 + \exp\{-(V + 27)/5\}}, \tag{24}$$

The coupling of metabolic resources ($O_2$) to $V$ is completed via the Hodgkin-Huxley formalism of a single neuron, where ion currents ($I_K$, $I_{Na}$, and $I_{Cl}$) influence the dynamics of $V$ obeying

$$C \frac{dV}{dt} = \left(-I_{Na} - I_K - I_{Cl} - I_{syn}\right), \tag{25}$$

where $C$ is the membrane capacitance, and $I_{syn}$ is the postsynaptic current from presynaptic neurons.

Wei et al.[16] also presented a minimal extension of the model to two coupled neurons, one excitatory and one inhibitory. Their aim was to model a phenomenon of excitatory-inhibitory interplay during seizures. They modeled synaptic coupling using a coupling scheme applicable to many neurons as given in ref. 70. Here, we extend the formalism to model networks of 400 neurons (320 excitatory and 80 inhibitory[71]), allowing us to explore the dynamics of small populations of neurons relevant to the generation of local field potentials. Each neuron receives input from ~80 randomly connected neurons (~64 excitatory and ~16 inhibitory); i.e., a connection probability of 0.2[72]. As shown in the Results, the model activity is self-sustaining even in the absence of external noise or external current.

The post synaptic current for a neuron ($I_{syn}$) is the sum of synaptic currents from $P$ presynaptic neurons[70,73,74], obeying

$$I_{syn} = \sum_{j=\{1,\ldots,P\}} G_{ex/inh}\left(V - E_{ex/inh}\right) S_j e^{-\frac{\chi_j}{3}}, \tag{26}$$

where $G_{ex}$ is the excitatory maximum synaptic conductance, $G_{inh}$ is the inhibitory maximum synaptic conductance, $E_{ex}$ is the reversal potential for excitatory synapses, $E_{inh}$ is the reversal potential for inhibitory synapses, and $S_j$ is the fraction of open receptors at the $j$th synapse, contributing to the overall synaptic conductance, modeled with first order kinetics[16,73,74] such that

$$\tau_{ex/inh} \frac{dS_j}{dt} = \frac{20}{1 + \exp\{-(V_j + 20)/3\}}(1 - S_j) - S_j, \tag{27}$$

where $\tau$ is the time constant for synaptic dynamics. The synapses can be either excitatory or inhibitory. For excitatory synapses $E_{ex} = 0$ mV, $\tau_{ex} = 4$ ms, and for inhibitory synapses $E_{inh} = -80$ mV, $\tau_{inh} = 8$ ms. Here $\chi_j$ models attenuation of the synapses when the presynaptic neuron is in the depolarization block[70], obeying

$$\frac{d\chi_j}{dt} = \eta(V_j + 50) - 0.4\chi_j, \tag{28}$$

where $\eta = 0.4$ when $-30$ mV $< V_j < -10$ mV and $\eta = 0$ otherwise.

As per the Results, we found values of $G_{ex} = 0.022$ mS/cm² and $G_{inh} = 0.374$ mS/cm² that generated self-sustaining asynchronous irregular (AI) states. It has previously been noted that networks of hundreds of neurons require ~10 times higher synaptic conductances than observed in real cells to exhibit self-sustained activity[18,20], which would otherwise require much larger networks (e.g., 16,000 neurons)[18]. Computational complexity of our model restricts us to work with small-scale networks.

Finally, for comparison with phenomena observed at the meso- and macroscale via LFP or EEG, we consider two measures of the summed network activity at time $t$. The first measure is the average firing rates ($\Phi^{fr}(t)$) of the excitatory population and inhibitory population[26], given by

$$\Phi^{fr}(t) = \frac{1}{N}\sum_{j=1}^{N}\phi_j^{fr}(t), \tag{29}$$

where $N$ is the total number of neurons, and $\phi_j^{fr}$ is the firing rate of the $j$th neuron. The second measure is the average of the post-synaptic currents ($\Phi^{syn}(t)$) of the excitatory neurons[26], given by

$$\Phi^{syn}(t) = \frac{1}{N_{ex}}\sum_{j=1}^{N_{ex}}I_{syn,j}(t), \tag{30}$$

where $N_{ex}$ is the number of excitatory neurons, and $I_{syn,j}(t)$ is the post-synaptic current (PSC) of the $j$th excitatory neuron at time $t$.

A 60 s simulation takes ~784 s on a Linux workstation with 3.7 GHz octa core processor. Parameter values and their descriptions are given in Table 1.

Numerical simulations were performed using the Runge-Kutta 4th order method with a time-step of 0.05 ms.

**Table 1 | Values and descriptions of model parameters**

| Parameter | Value | Description |
|---|---|---|
| $[Na^+]_{gi}$ | 18 mM | Intracellular Na concentration of glia |
| $\rho_{max}$ | 1.25 mM/s | Maximum pump rate |
| $\alpha$ | 5.3 g/mol | Conversion factor from pump current (mM/s) to $O_2$ consumption rate (mg/L/s) |
| $\lambda$ | 1 and 0.5 | Relative cell density for excitatory and inhibitory neurons |
| $\epsilon_o$ | 0.17 $s^{-1}$ | $O_2$ diffusion rate |
| $\gamma$ | 0.0445 (mM/s) /($\mu A/cm^2$) | Conversion factor from the current to concentration units |
| $\beta$ | 7 | Ratio of intracellular volume to extra-cellular volume |
| $\epsilon_k$ | 0.33 $s^{-1}$ | K diffusion rate |
| $G_{glia}$ | 8 mM/s | Glial uptake strength of potassium |
| $[Cl^-]_i$ | 6 mM | Intracellular Cl concentration |
| $[Cl^-]_o$ | 130 mM | Extracellular Cl concentration |
| $G_{Na}$ | 30 mS/$cm^2$ | Maximal conductance of Na current |
| $G_K$ | 25 mS/$cm^2$ | Maximal conductance of K current |
| $G_{NaL}$ | 0.0175 mS/$cm^2$ | Conductance of leak Na current |
| $G_{KL}$ | 0.05 mS/$cm^2$ | Conductance of leak K current |
| $G_{ClL}$ | 0.05 mS/$cm^2$ | Conductance of leak Cl current |
| $G_{ex}$ | 0.022 mS/$cm^2$ | Conductance of excitatory synapses |
| $G_{inh}$ | 0.374 mS/$cm^2$ | Conductance of inhibitory synapses |
| $E_{ex}$ | 0 mV | Reversal potential of excitatory synapses |
| $E_{inh}$ | −80 mV | Reversal potential of inhibitory synapses |
| $\tau_{ex}$ | 4 ms | Time constant of excitatory synapses |
| $\tau_{inh}$ | 8 ms | Time constant of inhibitory synapses |
| $C$ | 1 $\mu F/cm^2$ | Membrane capacitance |
| $N$ | 400 | Total number of neurons |
| $N_{ex}$ | 320 | Total number of excitatory neurons |
| $N_{in}$ | 80 | Total number of inhibitory neuronsinhibitory |
| $[K^+]_{Buffer}$ | 3.5 mM | K buffer concentration (normal value) |
| $[O_2]_{Buffer}$ | 32 mg/L | $O_2$ buffer concentration (normal value) |

## Coefficient of variation of inter-spike interval (CV$_{ISI}$) and pairwise correlation coefficient (CC)

The coefficient of variation of the inter-spike interval (CV$_{ISI}$) is defined as

$$CV_{ISI} = \left\langle \frac{\sigma_i^{ISI}}{\overline{ISI}_i} \right\rangle, \tag{31}$$

where $\langle\cdot\rangle$ denotes an average over all the neurons, and $\overline{ISI}_i$ and $\sigma_i^{ISI}$ are the mean and standard deviation, respectively, of the ISIs of neuron $i$.

The averaged pairwise cross-correlation (CC) between neurons in the network is given as[19]

$$CC = \left\langle \frac{\text{Cov}(n_i,n_j)}{\sigma(n_i)\sigma(n_j)} \right\rangle, \tag{32}$$

where $\langle\cdot\rangle$ indicates an average over all pairs of neurons, spike count $n_i$ is the number of spikes in sliding windows (non-overlapping 5 ms windows) of neuron $i$, Cov($n_i, n_j$) is the covariance between two spike counts $n_i$ and $n_j$, and $\sigma(n_i)$ is the standard deviation of neuron $i$'s spike counts.

## Kuramoto order parameter of synchronization, $R(t)$

To quantify synchronization, we used the Kuramoto order parameter $R(t)$ given by[75]

$$R(t) = \frac{1}{N}\left| \sum_{k=1}^{N} e^{i\phi_k(t)} \right|, \tag{33}$$

where $\phi_k(t)$ is the instantaneous phase calculated assuming linear phase (from 0 to $2\pi$) between two spikes such that the phase resets to 0 at every spike[76]; i.e.,

$$\phi_k(t) = 2\pi \frac{t - t_n^k}{t_{n+1}^k - t_n^k}, \tag{34}$$

where $t \in (t_n^k, t_{n+1}^k)$, and $t_n^k$ is the time of the $n$th spike in the $k$th neuron.

## Estimating asymmetry and sharpness of the average burst shape

To quantify the shapes of bursts, we averaged together burst time series for individual bursts, after rescaling to a common time axis. Asymmetry, $\Sigma$, is given by[8,77]

$$\Sigma(T) = \frac{\frac{1}{T}\int_0^T dt \langle y(t,T)\rangle (t-\bar{t})^3}{\left[\frac{1}{T}\int_0^T dt \langle y(t,T)\rangle (t-\bar{t})^2\right]^{3/2}}, \tag{35}$$

and sharpness, $K$, is given by,

$$K(T) = \frac{\frac{1}{T}\int_0^T dt \langle y(t,T)\rangle (t-\bar{t})^4}{\left[\frac{1}{T}\int_0^T dt \langle y(t,T)\rangle (t-\bar{t})^2\right]^2} - 3, \tag{36}$$

where $\langle y(t,T)\rangle$ is the average burst shape of duration $T$, $\bar{t} = \frac{1}{T}\int_0^T dt \langle y(t,T)\rangle t$, and we evaluate the integrals using the trapezoidal rule.

## Identification of different regimes in the $[K^+]_{Buffer}$-$[O_2]_{Buffer}$ plane

To identify states (Iso, AI, Bistable, BS, SZ) the average firing rate time series was estimated using a moving window of 500 ms duration. The state was defined as (i) isoelectric (Iso) if the average (across neurons) firing rate in each moving window is 0 spikes per 500 ms per cell; (ii) asynchronous-irregular (AI) if the average firing rate in each window is >0.75 spikes per 500 ms per cell; (iii) Bistable if both AI and Iso states are reachable depending on the initial conditions; and (iv) burst-suppression (BS) or seizure (SZ) if the average firing rate in any of

the moving windows is ≤0.75 spikes per 500 ms per cell. In particular, a seizure in this context is further defined by the sudden onset of high firing with marked synchronicity among neural spikes and sustained fluctuations in the firing rate time-series. With an increase in $[K^+]_{Buffer}$ the SZ regime transitions into BS. The transition point from SZ to BS was identified by visual inspection of the simulated firing rate time series, according to the presence or absence of bursts. These were identified as a sudden onset of high firing, followed by a decay to zero in the firing-rate time series.

### Estimation of trajectories of individual subject time-series in the $[K^+]_{Buffer}$-$[O_2]_{Buffer}$ plane

To test the validity of our model, we estimated model parameters that yield dynamics consistent with infant EEG during recovery from hypoxia. To do this, we analyzed scalp EEG recordings from 17 infants admitted to the tertiary level neonatal intensive care unit (NICU) in Helsinki University Central Hospital due to perinatal asphyxia (see Ref. [8] for details on this pre-existing dataset). The use of the retrospectively collected, archived patient data was approved by the Ethics Committee of the Hospital for Children and Adolescents, Helsinki University Central Hospital. Neurodevelopmental outcome categories were identified previously[8], such that good recovery (10 infants) denotes either normal development or only mild neuro-muscular disorders as assessed at their last visit to a routine neonatal outpatient clinic (age 12-39 months), while poor recovery (7 infants) denotes either death, severe neuromuscular disorders (severe injury), or a thalamic lesion. The EEG signals are collected in epochs from each infant in the NICU. The nature and duration of these epochs are at the discretion of the clinician in the NICU and are subject to interruption according to the clinical needs of each neonate. As such, the duration and timing available for analysis here vary across neonates. Each EEG recording (epochs) was represented using six features estimated using sequential non-overlapping windows. These features quantify the shapes of bursts and the distributions of their sizes and durations in two-channel infant EEG[8,78]. We computed the asymmetry (Σ) and sharpness (K) of the average burst shape computed from the bursts with duration (T) from 1280 ms to 5120 ms; i.e., duration bins representative of the characteristic average burst shape changes during pathological brain activity such as burst suppression[8,78]. For burst size and duration distributions, we calculated the width of the power-law scaling regime (given by the number of orders of magnitude, O) and the exponent (E) of the power law. For O we use the fitted range identified by the fitting of a strictly truncated power law distribution[79]. The slope of the fit is used as the exponent (E). Calculating O and E of fits to the burst area and burst duration distributions, respectively, yields four features.

We then estimated each infant's trajectory in the $[K^+]_{Buffer}$-$[O_2]_{Buffer}$ plane. Our approach uses a dynamic programming frame-work to minimize differences between EEG features captured from infants and those predicted by our model. We summarized the $i^{th}$ window from the $j^{th}$ epoch with a vector of these six features ($D_i^j$). Similarly, for each point in the $[K^+]_{Buffer}$-$[O_2]_{Buffer}$ plane, we calculate the same six features on the model time series of 5000 s, yielding $Q$ as an $N \times M \times 6$ array, where $N$ is the resolution of $[K^+]_{Buffer}$, and $M$ is the resolution of $[O_2]_{Buffer}$. Here, the burst extraction threshold for the firing rate time series was set at 0 spikes/s/cell, and for the post-synaptic current time series we select the threshold that maximizes the number of bursts[8].

**Constrained optimal projection of the data time-series on the model plane.** An optimal projection is the one that minimizes the projection score defined as the sum of the Euclidean distances ($\sum_i ||D_i - q_i||$) between the feature vectors of windows ($D_i$) and the feature vector of the corresponding entries ($q_i$) in the $[K^+]_{Buffer}$-$[O_2]_{Buffer}$ plane. An optimal projection is equivalent to a set of entries in the

$[K^+]_{Buffer}$-$[O_2]_{Buffer}$ plane $\langle q_1, \ldots, q_P \rangle$ such that

$$\langle q_1, \ldots, q_P \rangle = \operatorname*{argmin}_{\langle q_1, \ldots, q_P \rangle \in Q} \sum_{i=1}^{P} ||D_i - q_i||. \tag{37}$$

Next, we assume that biologically plausible trajectories are relatively smooth; i.e., consecutive epochs are near one another in the $[K^+]_{Buffer}$-$[O_2]_{Buffer}$ plane. We thus impose a constraint that the projections of two consecutive windows are within a radius of $R = 10$ apart in the $[K^+]_{Buffer}$-$[O_2]_{Buffer}$ plane.

We adopted a dynamic programming framework, a validated methodology for solving such non-trivial optimization problems[80], in Algorithm 1 to map each epoch onto the $[K^+]_{Buffer}$-$[O_2]_{Buffer}$ plane. In essence, the algorithm systematically searches the vast space of possible trajectories, keeping track of the projection scores for subsets of points in the trajectory. In detail, the information about the partial trajectories is stored in tables $s$ and $d$ of size $(P+1) \times (NM)$. Table $s$ stores the projection scores of the partial trajectories such that $s_{i+1,k}$ is the projection score of the optimal projection for $\langle D_1, \cdots, D_i \rangle$ when $D_i$ is projected onto the $k$th entry in $Q$ (step 6 in Algorithm 1). Therefore, the projection score of the partial optimal projection, $\langle q_1, \ldots, q_i \rangle$, is the minimum value in the $(i+1)$th row of table $s$. Table $d$ stores the indices of the partial projections. For a partial optimal projection $\langle q_1, \ldots, q_i \rangle$, if the index of $q_i$ is $k$, then $d_{i+1,k}$ contains the index of $q_{i-1}$. This way the optimal projection can be traced from the table $d$ (step 19 to 21 in Algorithm 1).

**Algorithm 1.** Dynamic programming algorithm for finding constrained optimal projection

> **Data:**
> $s$ - table of size $(P+1) \times (NM)$ for storing partial projection scores
> $d$ - table of size $(P+1) \times (NM)$ for tracking the optimal projection
> 1:  **function** FindConstrainedOptimalProjection($Q, \langle D_1, \cdots, D_P \rangle, R$)
> ▷ Input: $Q$ − model plane of size $M \times N$,
> $\langle D_1, \cdots, D_P \rangle$ − data time-series,
> $R$ − radius for constraint
> ▷ Output: $\langle z_1, \ldots, z_P \rangle$ − indices of the optimal projection
> 2: Initialize $1^{st}$ row of $s$ and $d$ to 0
> 3: **for** $i \leftarrow 1$ to $P$ **do**
> 4: **for** $k \leftarrow 1$ to $NM$ **do**
> 5: $(value, index) \leftarrow$ ConstrainedMin($s_i, k, R$)
> 6: $s_{i+1,k} \leftarrow ||D_i - Q_k|| + value$
> 7: $d_{i+1,k} \leftarrow index$
> 8: **end for**
> 9: **end for**
> 10: **return** TraceIndexOfOptimalProjection($d, s_{P+1}$)
> 11:  **end function**
>
> 12:  **function** ConstrainedMin($s_i, k, R$)
> 13: Finds the minimum $value$ of partial score ($s_i$) and its $index$ in the vicinity (defined by $R$) of index $k$
> 14: **return** ($value, index$)
> 15:  **end function**
>
> 16:  **function** TraceIndexOfOptimalProjection($d, s_P$)
> 17: $z$ - array of size $P$ for storing the indices of the optimal projection
> 18: $z_P \leftarrow$ argmin($s_p$)
> 19: **for** $i \leftarrow (P-1)$ to 1 **do**
> 20: $z_i \leftarrow d_{i+2, z_{i+1}}$
> 21: **end for**
> 22: **return** $z$
> 23:  **end function**

### Reporting summary

Further information on research design is available in the Nature Portfolio Reporting Summary linked to this article.

## Data availability

The raw and processed simulation data generated in this study, which are plotted in the figures, have been deposited in the Figshare database, accessible via https://doi.org/10.6084/m9.figshare.23514531.v1. The EEG data from human infants are sensitive data that cannot be distributed without pertinent preprocessing to ensure anonymity as well as relevant data sharing agreements with Helsinki University Hospital (via author S.V.). However, the anonymized analytic derivative of this EEG data (EEG power, such as in Fig. 2a–d) has been deposited in the Figshare database, accessible via the same https://doi.org/10.6084/m9.figshare.23514531.v1.

## Code availability

MATLAB code is accessible via https://doi.org/10.5281/zenodo.8013692.

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

## Acknowledgements

This work was supported by the National Health and Medical Research Council (Project Grants 1145168 to J.A.R. and 1144936 to J.A.R., and Investigator Grant 2008612 to M.B.), Finnish Academy (#1332017 to S.V.), and Juselius Foundation (to S.V.). We also thank Anirudh N. Vattikonda for helpful comments and discussions.

## Author contributions

S.D., M.B., and J.A.R. designed the study. S.D. performed the simulations. S.D., M.B., and J.A.R. performed the analyses. K.K.I. and S.V.

provided the infant EEG data. All authors discussed the results and contributed to writing the paper.

## Competing interests

S.V., M.B., and J.A.R. hold a licensed patent on the burst metrics used in this paper. S.D. and K.K.I. declare no competing interests.

## Ethics

The use of the retrospectively collected, archived patient data was approved by the Ethics Committee of the Hospital for Children and Adolescents, Helsinki University Central Hospital.
