## [Peer Review File · Nature Communications]

Mechanisms underlying pathological cortical bursts during metabolic depletionREVIEWER COMMENTS

Reviewer #1 (Remarks to the Author):

The authors built a computational network model consists of 320 excitatory and 80 inhibitory neurons with the dynamics of oxygen, potassium and sodium concentration. They found that oxygen depletion drives transition from normal activity to several pathological activity patterns (isoelectric, burst suppression, and seizure). They examined the trajectories through parameter space that correspond to good recovery and bad recovery, and validate the model against the clinical scalp EEG from 17 infants following birth asphyxia. The authors suggested that these findings may offer new insights into the coupling between neuronal and energy activity more generally.

Overall, the paper is well organized. However, I have some comments.

Critique:

Why did the authors use $[K^+]_{buffer}$ as a proxy for energy demand? Energy demand should be dynamically changed based on the neuronal activities, not $[K^+]_{buffer}$. Also, how can increasing energy demand ($[K^+]_{buffer}$) is essential for recovery?

Is there any experimental evidence observed as similar as shown in Fig4?

In Fig7, is there any way to quantify the difference between good recovery and bad recovery? It's hard to tell the difference based on the trajectories.

Are these six features enough to capture the characteristic of seizures?

Minor:

Fig 1, missing the scale of x and y axis.

How long does it take to compute 60s simulation?

Fig 2 a-d, why did the authors use different duration of the EEG signals?

What is membrane potential for a single neuron activity look like in each state (AI, BS, SZ, Iso, Bistable) in the model? How did the authors define these states (AI, BS, SZ, Iso, Bistable)?

In the text, the authors use $\phi^{fr}(t)$ and $\phi^{syn}(t)$, in the y-label of Fig 3b and c, the authors use F.R. and PSC. Please keep them consistent. The same issue should also be addressed in the description of Fig 5bc, and Fig6bc.

Reviewer #2 (Remarks to the Author):

The paper by Dutta and colleagues studies potential mechanisms associated with aberrant brain activity in the context of metabolic injury in neonates. Using a computational modeling approach, the authors study the ways in which neuronal and metabolic processes may interact to give rise to particular activity patterns, such as burst suppression, in situations of metabolic distress. In this context, the differential role of Oxygen and Potassium supply are examined and found to play key roles in how the brain may transition to and from different regimes of activity. Finally, the modeling results are used to perform a type of state estimation, wherein the above parameters are inferred from actual patient electrophysiological data. It is shown that the inferred parameters are compatible with patient outcomes and type of pathology.

Overall, this is a well-written paper and the premise is clear. The computational modeling approach, based on voltage-gated conductance models and biophysically-grounded metabolic equations, is generally sound. The question of how metabolic processes interact with neuronal dynamics is, as stated by the authors, is an important and often neglected one. Thus, the approach here is a step in the right direction. I do, however, have several questions that I feel need resolution in order for the paper to meet its stated impacts.

Primary comments (in order of appearance):

- Line 157. The notion of bistability in this context is confusing. I interpret this to mean that the nominal AI regime, i.e., as depicted in Figure 3, corresponds to some sort of limit cycle attractor in high-dimensional space, which co-exists with another asymptotically stable fixed point corresponds on the isoelectric regime. If this is correct, it would suggest that the healthy brain could, at any time, exist the basin of attractor associated with the former and become isoelectric. Is this the correct interpretation of the dynamics as presented? If so, I find this to be a potentially problematic premise, unless an argument can be made that the basin of attraction associated with the isoelectric state is not in a physiologically-relevant range (Line 163-166). However, the assertion that isoelectricity can arise for certain initial conditions calls this into question. Could the authors comment/clarify?
 - Line 204-207 and Figure 5: It would be good to interpret these mechanisms a bit more from a dynamical systems perspective. It would seem that there are actually three time-scales manifesting in this regime. What is the mechanism of the shorter and faster LFP bursts? Do those also appear in the baseline AI state, or is this a new regime that arises in the low O2 context? Could the authors attempt a bifurcation analysis of sorts (even numerically or approximately) to understand these dynamics?
 - Section at Line 242: Similar to above, some insight would be useful here as to understand what is going on—it would seem there is some sort of hysteresis effect wherein decreasing and then increasing O2 lead to asymmetric expression of regimes. Certainly, this seems compatible with clinical observations. What is mediating this effect in the model? A dynamical-mechanistic interpretation here would, I feel, bolster the subsequent observations regarding the role of K⁺ in the model.
 - Line 285-299: I support the premise of using models of this sort to facilitate latent state estimation, and the results here, even as proof of concept, are a nice addition to the paper. Some robustness analysis and methodological clarification, however, would be quite useful. For example, the authors provide two frameworks for deriving simulated LFP signals (i.e., (30) and (31)). Are the results robust to the use of either framework? It might surmise, e.g., that the shape parameters have some sensitivity to the choice of LFP surrogate. Similarly, how do the results hold up as parameters are removed (i.e., are all 6 parameters necessary, or are some more important than others)?
 - Paragraph starting with line 300: Another sanity check type of analysis that would be useful would be to infer back the latent parameters from the simulated time series, in order to make sure there is a near 1-1 correspondence. I surmise that the continuity regularizer within the optimization problem in Algorithm 1 plays a key role here, since I imagine there may be many points in the latent space associated with similar values of the 6 parameters.
 - In general, Algorithm 1 looks like a lot of machinery for what is in essence a least squares problem, and it would be preferable to use a validated methodology in this context. Can the authors comment on the need for a newly developed algorithm here?
- Minor comments:
- Lines 140-142: Can the authors comment on the mechanism of these 26Hz oscillations? Are these mediated by the kinetics of inhibition/IPSCs in the network?
 - I would prefer that code be made available through a publicly accessible repository, such as GitHub, rather than by request on the authors'

We thank the reviewers for their comments. Our responses (black) and changes to the text (red) are interleaved with the comments (italic).

Reviewer #1:

***RI.1:** The authors built a computational network model consists of 320 excitatory and 80 inhibitory neurons with the dynamics of oxygen, potassium and sodium concentration. They found that oxygen depletion drives transition from normal activity to several pathological activity patterns (isoelectric, burst suppression, and seizure). They examined the trajectories through parameter space that correspond to good recovery and bad recovery, and validate the model against the clinical scalp EEG from 17 infants following birth asphyxia. The authors suggested that these findings may offer new insights into the coupling between neuronal and energy activity more generally.*

Overall, the paper is well organized. However, I have some comments.

We thank the reviewer for their appraisal and constructive comments.

***RI.1:** Critique:*

Why did the authors use $[K^+]_{\text{buffer}}$ as a proxy for energy demand? Energy demand should be dynamically changed based on the neuronal activities, not $[K^+]_{\text{buffer}}$.

A1.1:

We agree with the reviewer that energy demand is not solely dependent on $[K^+]_{\text{Buffer}}$ but also on dynamical changes in neural activity through ion dynamics, and in particular the Na-K pump activity. In the revised manuscript, we no longer refer to the buffers as proxies for energy demand and supply. Instead, we refer to $[K^+]_{\text{Buffer}}$ as potassium supply and $[O_2]_{\text{Buffer}}$ as oxygen supply, and are more explicit about how these act as control parameters modulating the extracellular concentrations $[K^+]_o$ and $[O_2]_o$.

Moreover, to better emphasize the relationships between supply variables and neuronal variables, we have moved the model schematics from the supplementary figures into the main text as new panel i in Fig. 1.

Abstract (p1):

We find that restricting oxygen supply drives transitions from normal activity to several pathological activity patterns (isoelectric, burst suppression, and seizures), depending on the potassium supply.

Results (p2 and p3):

We investigate the parameter space of external potassium and oxygen supply to understand the effect of metabolism on neural activity. The oxygen supply from the blood vessels is modeled as diffusion from a reservoir of concentration $[O_2]_{\text{Buffer}}$ [16]. Similarly, the diffusion of potassium into the extracellular space from blood vessels and other sources is modeled by the concentration ($[K^+]_{\text{Buffer}}$) of potassium in the reservoir [16, 21]. Previous research has identified $[O_2]_{\text{Buffer}}$ and $[K^+]_{\text{Buffer}}$ as important control parameters for single-neuron dynamics and experimental preparations [16, 17, 22].

The activity of the energy-consuming Na^+K^+ pumps (Eq. 4) is influenced by several factors (Fig. 1i). The extracellular potassium ($[K^+]_o$), intracellular sodium ($[Na^+]_i$), and extracellular oxygen ($[O_2]_o$) concentrations directly impact the pump currents (eqs. 1 & 2). Meanwhile, $[O_2]_{\text{Buffer}}$ and $[K^+]_{\text{Buffer}}$ set the baseline values for $[O_2]_o$ (eqs. 3 & 4) and $[K^+]_o$ (eqs. 1, 2, & 5), respectively. As a result, an

increase in $[K^+]_{\text{Buffer}}$ indirectly increases the baseline energy demand by raising the equilibrium value of $[K^+]_o$.

Methods (p23):

All else being equal, increasing $[K^+]_{\text{Buffer}}$ increases $[K^+]_o$ (Eq. 5), which increases the neural and glial $\text{Na}^+\text{-K}^+$ pump currents (Eqs. 1 and 2). This in turn increases O_2 consumption (Eq. 4); i.e., increases in $[K^+]_{\text{Buffer}}$ increase the baseline energy demand on top of which are superimposed the dynamically changing demands, which derive from neural firing.

Fig. 1i. Model schematics. We use a network of 400 neurons (320 excitatory and 80 inhibitory) such that each neuron receives synaptic inputs (S_j) from 80 random neurons. The dendritic summation of these inputs results in the postsynaptic current (I_{syn}) for the neuron. I_{syn} modulates the neuron's membrane potential (V). V is also modulated by the intrinsic ion currents (I_K and I_{Na}), which result from the net ion flow between the intracellular and extracellular spaces. Intracellular and extracellular ion concentrations ($[K^+]$ and $[Na^+]$) establish gradients across the membrane (reversal potentials E_K and E_{Na}). Ion pumps (I_{pump} and I_{gliapump}) modulate ion concentrations to maintain concentration gradients, expending energy derived from O_2 bonds [23]. The extracellular concentration of O_2 is mediated by $[O_2]_{\text{Buffer}}$. The model also incorporates diffusion of potassium from distal sources parameterized by $[K^+]_{\text{Buffer}}$.

RI.2: Also, how can increasing energy demand ($[K^+]_{\text{buffer}}$) be essential for recovery?

A1.2: We have re-written the Results section describing the role of K^+ in post-hypoxic recovery using only descriptive language and avoiding strong statements such as K^+ being “essential” for recovery. The rewritten subsections (p12-16) now have headings “Role of potassium in recovery from hypoxic insult”, “Conditions for recovery at normal $[K^+]_{\text{Buffer}}$ ”, “Conditions for recovery via high $[K^+]_{\text{Buffer}}$ ”, “Recovery from hypoxia via BS at high $[K^+]_{\text{Buffer}}$ ”, and “Mechanisms for the role of K^+ in successful recovery”.

We now also address this in Discussion (p20):

Our exploration of recovery trajectories revealed that in addition to timely re-oxygenation, an increase in $[K^+]_{\text{Buffer}}$ facilitates the restoration of healthy dynamics by preventing the over-correction of $[K^+]_o$ during re-oxygenation. This suggests that a substantial increase in potassium following hypoxia (as

observed empirically [31-42,51]) could be a protective mechanism that brings the dynamics closer to the BS regime.

RI.3: *Is there any experimental evidence observed as similar as shown in Fig4?*

A1.3: We are not aware of experiments observing changes in neural activity for multiple controlled values of $[K^+]_{\text{Buffer}}$ and $[O_2]_{\text{Buffer}}$. We have edited the text to acknowledge and contextualize this in the Discussion (**p21**),

Validation of model predictions and calibration of model parameters can be achieved through future experiments. Although we are not aware of experiments that have investigated changes in neural activity across multiple controlled values of $[K^+]_{\text{Buffer}}$ and $[O_2]_{\text{Buffer}}$, independent evidence supports the emergence of seizure-like activity for higher $[K^+]_{\text{Buffer}}$ [60,61], and conversely, for lower $[O_2]_{\text{Buffer}}$ [22]. Moreover, substantial evidence suggests that hypoxia increases extracellular potassium [9, 31-43, 62-67], thus supporting our fundamental approach that understanding the response to hypoxia activity requires a joint modeling of both $[K^+]_{\text{Buffer}}$ and $[O_2]_{\text{Buffer}}$. More broadly, we suggest that systematically mapping out the $[K^+]_{\text{Buffer}}$ - $[O_2]_{\text{Buffer}}$ plane in experimental systems could prove fruitful in understanding dynamics under metabolically challenged conditions and provide independent validation of our model.

RI.4: *In Fig7, is there any way to quantify the difference between good recovery and bad recovery? It's hard to tell the difference based on the trajectories.*

A1.4: We performed new analyses, reporting that the mean $\Delta[K^+]_{\text{Buffer}}$ captures the difference in trajectories associated with good versus poor recovery, quantifying our prior qualitative interpretation of the trajectories in Fig. 7. We have added this new result to the main text and appended a new figure to the previous Fig 7i,j as a separate Fig 8:

Results (p17):

Quantitatively, the mean $\Delta[K^+]_{\text{Buffer}}$ of the last epoch of babies with good outcome was significantly lower than that of babies with poor outcome (two-tailed t-test $p = 0.0077$, t-statistic = - 3.0756, $df = 14.9771$, Fig. 8c).

Fig 8: Inferred trajectories in the model parameter space from neonatal data. a, Neonates with good recovery outcome. **b**, Neonates with poor recovery outcome. Colors denote time along the complete trajectories, from blue to red, with black dashed lines connecting epochs. Left panels show trajectories from an exemplar baby in each group. Right panels show the median trajectory for each group. **c**, Box and whisker plots for $\Delta[K^+]_{\text{Buffer}}$ from the last epoch for infants with good versus poor recovery outcomes. Circles, all individual (per infant) $\Delta[K^+]_{\text{Buffer}}$ values; center line, median; box limits, upper and lower quartiles; whiskers, minimum and maximum.

RI.5: Are these six features enough to capture the characteristic of seizures?

A1.5: We have included a new paragraph in the discussion section (p21):

Seizures and BS both pose significant metabolic challenges to the cortex and can occur together after birth hypoxia. Our model can qualitatively reproduce essential seizure properties, including high-amplitude waveforms, variable durations, oscillations, and slowing of frequency harmonics. For quantitative model parameter estimation, it remains an open question whether the six BS features used in this study capture clinically-meaningful properties of seizures. While it is likely that the six features capture some fundamental characteristics such as duration, we expect that the oscillatory nature of seizures contains unique information that would be best captured by measures sensitive to the fundamental frequency, harmonics, and gradual intra-ictal slowing (chirps), which will be explored in future work.

RI.6: Minor:

Fig 1, missing the scale of x and y axis.

A1.6: We have added the scales.

RI.7: *How long does it take to compute 60s simulation?*

A1.7: We have added a note to the Methods (**p26**):

A 60 s simulation takes ~784 s on a Linux workstation with 3.7 GHz octa core processor.

RI.8: *Fig 2 a-d, why did the authors use different duration of the EEG signals?*

A1.8: We now clarify this in Methods (**p29**):

The EEG signals are collected in epochs from each infant in the NICU. The nature and duration of these epochs are at the discretion of the clinician in the NICU and subject to interruption according to the clinical needs of each neonate. As such, the duration and timing available for analysis here vary across neonates.

RI.9: *What is membrane potential for a single neuron activity look like in each state (AI, BS, SZ, Iso, Bistable) in the model?*

A1.9: We have added a new Supplementary Figure S2 (introduced on **p9**):

We show exemplar single-neuron membrane potential time series for all states in Supp. Fig. S2.

Supplementary Figure S2. Example membrane potential of a single excitatory neuron (black) and single inhibitory neuron (red) for different states. a, Isoelectric. b, Bistable Asynchronous Irregular. c, High $[K^+]_{\text{Buffer}}$ Asynchronous Irregular. d, Burst Suppression. e, Seizure.

RI.10: How did the authors define these states (AI, BS, SZ, Iso, Bistable)?

A1.10: We have added the following explanation to the Methods as a new subsection “Identification of different regimes in the $[K^+]_{\text{Buffer}}$ - $[O_2]_{\text{Buffer}}$ plane” (p29):

To identify states (Iso, AI, Bistable, BS, SZ) the average firing rate time series was estimated using a moving window of 500 ms duration. The state was defined as (i) isoelectric (Iso) if the average (across neurons) firing rate in each moving window is 0 spikes per 500 ms per cell; (ii) asynchronous-irregular (AI) if the average firing rate in each window is greater than 0.75 spikes per 500 ms per cell; (iii) Bistable if both AI and Iso states are reachable depending on the initial conditions; and (iv) burst-suppression (BS) or seizure (SZ) if the average firing rate in any of the moving windows is less than or equal to 0.75 spikes per 500 ms per cell. With an increase in $[K^+]_{\text{Buffer}}$ the SZ regime transitions into BS. The transition point from SZ to BS was identified by visual inspection of the simulated firing rate time series, according to the presence or absence of bursts. These were identified as a sudden onset of high firing, followed by a decay to zero in the firing-rate time series. Seizures display sudden onset of

high firing with marked synchronicity among neural spikes and sustained fluctuations in the firing rate time-series.

RI.11: *In the text, the authors use $\hat{\phi}^{fr}(t)$ and $\hat{\phi}^{syn}(t)$, in the ylabel of Fig 3b and c, the authors use F.R. and PSC. Please keep them consistent. The same issue should also be addressed in the description of Fig 5bc, and Fig6bc.*

A1.11: We have harmonized the labeling.

Reviewer #2:

The paper by Dutta and colleagues studies potential mechanisms associated with aberrant brain activity in the context of metabolic injury in neonates. Using a computational modeling approach, the authors study the ways in which neuronal and metabolic processes may interact to give rise to particular activity patterns, such as burst suppression, in situations of metabolic distress. In this context, the differential role of Oxygen and Potassium supply are examined and found to play key roles in how the brain may transition to and from different regimes of activity. Finally, the modeling results are used to perform a type of state estimation, wherein the above parameters are inferred from actual patient electrophysiological data. It is shown that the inferred parameters are compatible with patient outcomes and type of pathology.

Overall, this is a well-written paper and the premise is clear. The computational modeling approach, based on voltage-gated conductance models and biophysically-grounded metabolic equations, is generally sound. The question of how metabolic processes interact with neuronal dynamics is, as stated by the authors, is an important and often neglected one. Thus, the approach here is a step in the right direction. I do, however, have several questions that I feel need resolution in order for the paper to meet its stated impacts.

We thank the reviewer for their positive appraisal and constructive comments.

R2.1: Primary comments (in order of appearance):

- Line 157. The notion of bistability in this context is confusing. I interpret this to mean that the nominal AI regime, i.e., as depicted in Figure 3, corresponds to some sort of limit cycle attractor in high-dimensional space, which co-exists with another asymptotically stable fixed point corresponds on the isoelectric regime. If this is correct, it would suggest that the healthy brain could, at any time, exist the basin of attractor associated with the former and become isoelectric. Is this the correct interpretation of the dynamics as presented? If so, I find this to be a potentially problematic premise, unless an argument can be made that the basin of attraction associated with the isoelectric state is not in a physiologically-relevant range (Line 163-166). However, the assertion that isoelectricity can arise for certain initial conditions calls this into question. Could the authors comment/clarify?

A2.1: The role and dynamic stability of the isoelectric state is an important consideration that we have now addressed with additional analyses. We included two supplementary figures and corresponding new text:

Results (p8):

Note that these conditions arise in the absence of an external stochastic drive. To assess the stability of the isoelectric state, we performed additional simulations in the setting where the network receives external stochastic drive. We find that the coexisting isoelectric state is only stable (non-spiking) in the presence of very small perturbations ($\lesssim 1.5 \mu\text{A}/\text{cm}^2$ amplitude; Supp Fig. S3a). External noise of greater amplitude shifts the dynamics to the AI regime (Supp. Fig. S3b-d). In contrast, under normal physiological conditions in the AI state, the amplitudes of spontaneous post-synaptic currents are approximately $50\text{-}100 \mu\text{A}/\text{cm}^2$ (Supp. Fig. S4, left panels), hence up to 2 orders of magnitude stronger than required to disrupt the isoelectric state (Supp. Fig. S4, right panels). Therefore, while the isoelectric state co-exists as an attractor in this region of parameter space, it has a very small basin of attraction and as such is unlikely to be observed under normal physiological conditions.

Supplementary Figure 3. Stochastic noise transitions the bistable regime from isoelectric to AI. **a**, Starting from initial conditions in the isoelectric state, a weak stochastic input current of amplitude $\sim 1.5 \mu\text{A}/\text{cm}^2$ (red asterisk), or higher, yields spiking dynamics. **b-d**, Further analyses show that the mean E-I balance (**b**), coefficient of variation (**c**), and correlation coefficient (**d**) rapidly converge to values consistent with the noise free AI state.

Supplementary Figure 4. Comparison of representative post-synaptic currents (Φ^{syn} , left panels) and external stochastic currents (I_{noise} , right panels) sufficient to perturb the system away from the isoelectric state during physiological values of $[\text{K}^+]_{\text{Buffer}}$ and $[\text{O}_2]_{\text{Buffer}}$. Each row represents an excitatory neuron picked at random: Excitatory neurons numbered 11, 258, 215, 244, and 88. The inset in the right panel is the zoom showing 5 s of I_{noise} for excitatory neuron number 258.

R2.2: - Line 204-207 and Figure 5: It would be good to interpret these mechanisms a bit more from a dynamical systems perspective. It would seem that there are actually three time-scales manifesting in this regime. What is the mechanism of the shorter and faster LFP bursts? Do those also appear in the baseline AI state, or is this a new regime that arises in the low O_2 context? Could the authors attempt a bifurcation analysis of sorts (even numerically or approximately) to understand these dynamics?

A2.2: We agree there appear to be three important time scales and have added the following new analysis with a supplementary figure and text to the manuscript:

Results (p10) :

These BS dynamics derive from the complex interplay of neuronal and network dynamics across three distinct time scales (Supp. Fig. S5). The fast time scale of individual spikes (V) derives from the classic Hodgkin-Huxley-type membrane capacitance (Supp. Fig. S5b). The second time scale corresponds to the repetitive firing of many cells within a burst timelocked to the recovery dynamics of potassium (Supp. Fig. S5c). The third time scale reflects the interplay of slow metabolic ($[\text{O}_2]_o$, Supp. Fig. S5d) and ionic ($[\text{Na}^+]_i$, Supp. Fig. S5e) processes, yielding the duration of the bursts and the interval between them. The second and third time scales emerge because the ionic concentrations change in response to neuronal firing. The characteristics of the network bursts (such as size and duration) depend on the number of recruited neurons and the timing of the onsets of the bursts in relationship to the recovery of $[\text{O}_2]_o$ and $[\text{Na}^+]_i$: As the number of recruited neurons varies within each network burst, the system fluctuates between small (few neurons recruited) and large bursts (most neurons recruited). Small bursts typically occur when a burst is initiated when the metabolic states of most of the system's neurons are still recovering from the previous burst (see example in Supp. Fig. S5 at ~ 150 s). Conversely, larger bursts occur when $[\text{O}_2]_o$ and $[\text{Na}^+]_i$ have recovered in most neurons (see example in Supp. Fig. S5 at

~100 sec). The time scales associated with BS bursts are longer than those of the activity fluctuations in the AI state, which do not exhibit large, network-wide changes in ionic concentrations and thus do not engage these slower time scales.

Supplementary Figure S5: Individual state variables of all neurons (left: excitatory; right: inhibitory) during the BS state.

We note that the overarching objective of the study is to understand clinical neurophysiological phenomena using this neuron-metabolic model. As such we feel that a formal bifurcation analysis (a nontrivial undertaking in a complex high dimensional system) is beyond the scope of the present study.

R2.3: - Section at Line 242: Similar to above, some insight would be useful here as to understand what is going on—it would seem there is some sort of hysteresis effect wherein decreasing and then increasing O_2 lead to asymmetric expression of regimes. Certainly, this seems compatible with clinical observations. What is mediating this effect in the model? A dynamical-mechanistic interpretation here would, I feel, bolster the subsequent observations regarding the role of K^+ in the model.

A2.3: We performed additional analyses to compare the dynamics of the two scenarios presented in Fig. 7b. These indeed give greater insight into the role of K^+ in the recovery simulations presented in the paper with a numerically-based dynamic insight. These are included in a new subsection titled “Mechanisms for the role of K^+ in successful recovery” on **p16**.

Mechanisms for the role of K^+ in successful recovery

In sum, we observe an apparently protective role of increased K^+ (whether or not BS is involved). To gain an understanding of this, we performed numerical simulations (Fig. 7i-k) straddling the maximum survivable duration of hypoxia, which is in between the hypoxia duration of 1 s (Fig. 7b, top) and 2 s (Fig. 7b, bottom). During the period of hypoxia, $[O_2]_o$ decreases, and $[K^+]_o$ and $[Na^+]_i$ increase due to the impact of low extracellular oxygen on the Na^+-K^+ pumps. When the hypoxia ends, $[O_2]_o$ slowly recovers allowing Na^+-K^+ pumps to restore $[Na^+]_i$ and $[K^+]_o$. Irrespective of whether activity persists or ceases, $[K^+]_o$ returns to its pre-hypoxia values prior to the return of $[Na^+]_i$ because of the differences in their respective time scale parameters. Thereafter, $[K^+]_o$ continues to decrease below its pre-hypoxia range. This is because as the Na^+-K^+ pump continues to restore $[Na^+]_i$, it exchanges 3 Na^+ for 2 K^+ , decreasing $[K^+]_o$ below its equilibrium value. This decreases the reversal potential of K^+ (E_K), and therefore also decreases the resting membrane potential. Numerical simulations (Fig. 7i-k) suggest that if the over-correction of K^+ is too large, neural activity in the system exhibits the delayed “collapse”—that is, it suddenly converges to the isoelectric state. Conversely, if the hypoxia is sufficiently brief so that sodium recovers before any overcorrection of K^+ , the system remains in the AI state. The critical value of K^+ —the separatrix—appears to be approximately 4 mM.

Fig 7i-k, Termination of activity in panel b due to over-correction of K^+ . Multiple simulations between hypoxia (onset denoted by vertical black lines) of duration 1 s (cf. panel b, top row) and 2 s (cf. panel b, bottom row). Simulations where activity survived are shown in green, while simulations where activity ceased are shown in red. **i**, Mean $[K^+]_o$ across neurons. **j**, Mean $[Na^+]_i$ across neurons. **k**, Mean $[O_2]_o$ across neurons.

R2.4: - Line 285-299: I support the premise of using models of this sort to facilitate latent state estimation, and the results here, even as proof of concept, are a nice addition to the paper. Some robustness analysis and methodological clarification, however, would be quite useful. For example, the authors provide two frameworks for deriving simulated LFP signals (i.e., (30) and (31)). Are the results

robust to the use of either framework? It might surmise, e.g., that the shape parameters have some sensitivity to the choice of LFP surrogate.

A2.4: As suggested, we repeated the infant-specific trajectory analysis using the PSC power and have added the analyses in the main text alongside three new supplementary figures:

Results (**p17**):

By repeating these analyses using the instantaneous power time series of Φ^{syn} (i.e., the square of the absolute values of the Hilbert-transform-derived analytical signal), we obtain trajectories that broadly resemble those derived from the Φ^{tr} time series. However, burst metrics are sensitive to the choice of the LFP proxy, resulting in some differences between the two sets of trajectories (Supp. Figs. S7 & S8). Nevertheless, the inferred changes in potassium levels differentiating good versus poor outcomes remain largely preserved (Supp. Fig. S9).

And added a note on proper forward modelling to the Discussion (**p20**):

Moreover, comparison with EEG (and other modalities) would be improved using a detailed forward (observation) model that maps neuronal variables to measured quantities (e.g. taking into account electrode geometry and tissue properties).

Supplementary Figure S7: Comparison between metabolic parameters inferred from the firing rate time series (original, black) and from the instantaneous power of postsynaptic current time series (red) for infants with poor recovery outcomes.

Supplementary Figure S8: Comparison between metabolic parameters inferred from the firing rate time series (original, black) and from the instantaneous power of postsynaptic current time series (red) for infants with good recovery outcomes.

Supplementary Figure S9. Box and whisker plots for $\Delta[K^+]_{\text{Buffer}}$ from the last epoch for infants with good recovery outcome versus poor recovery outcome. a, Using firing-rate time series as the LFP proxy. b, Using instantaneous power of the postsynaptic currents time series as the LFP proxy. Circles, all individual (per infant) $\Delta[K^+]_{\text{Buffer}}$ values; center line, median; box limits, upper and lower quartiles; whiskers, minimum and maximum.

R2.5: Similarly, how do the results hold up as parameters are removed (i.e., are all 6 parameters necessary, or are some more important than others)?

A2.5: We have performed the suggested analysis: we reconstructed the inferred trajectories by removing one parameter each time, while keeping the other 5 parameters. We have added these analyses in the main text alongside three new supplementary figures.

Results (p17):

In addition, we examined potential redundancies between burst metrics by removing one parameter at a time and reconstructing the inferred trajectories using the remaining five metrics. We found that these five-parameter trajectories are broadly similar to their original six-parameter trajectories, implying partial redundancy, though there is no universally redundant parameter (Supp. Figs. 10 & 11).

Discussion (p21),

Future work could also use machine learning tools to identify optimal personalized or global sets of burst metrics.

Supplementary Figure 10: Comparing the inferred parameters using all six burst metrics (original) with scenarios where one metric is deleted while keeping the other five, for infants with poor recovery outcomes.

Supplementary Figure 11: Comparing the inferred parameters using all six burst metrics (original) with scenarios where one metric is deleted while keeping the other five, for infants with good recovery outcomes.

R2.6: - *Paragraph starting with line 300: Another sanity check type of analysis that would be useful would be to infer back the latent parameters from the simulated time series, in order to make sure there is a near 1-1 correspondence. I surmise that the continuity regularizer within the optimization problem in Algorithm 1 plays a key role here, since I imagine there may be many points in the latent space associated with similar values of the 6 parameters.*

A2.6: We agree that a sanity check of this nature is instructive. To address this, we now infer the latent parameters from three distinct synthetic trajectories, each containing eight points. We have included the results of this analysis in the main text.

Results (p18):

To assess the validity of the trajectory inference Algorithm 1, we inferred metabolic parameters from three distinct synthetic trajectories: a straight line, a kinked trajectory, and a loop (Supp. Fig. S12a). We then estimated burst statistics from the simulated time series and used the same algorithm employed for the empirical data to infer the optimal parameter trajectory. We found that the inferred parameters captured the basic trends present in the ground truth synthetic trajectories and are distinguishable from one another (Supp. Fig. S12b). Notably, the straight and kinked trajectories can be easily disambiguated from the looped curve, an important property observed in the good (straighter) versus poor (more looped) outcome neonates.

Supplementary Figure 12: Inferring back the parameters from simulated time-series: a, Three distinguishable synthetic trajectories. **b**, Inferred parameters from the simulated time series of the trajectories in panel **a**.

R2.7: - In general, Algorithm 1 looks like a lot of machinery for what is in essence a least squares problem, and it would be preferable to use a validated methodology in this context. Can the authors comment on the need for a newly developed algorithm here?

A2.7:

We did not develop the dynamic programming methodology employed in the paper, but merely adopted a previously validated methodology of dynamic programming for solving non-trivial optimization problems. We apologize for the lack of clarity in the submitted manuscript and have edited the text as follows,

Methods (p30):

We adopted a dynamic programming framework, a validated methodology for solving such non-trivial optimization problems [81], in Algorithm 1 to map each epoch onto the $[K^+]_{\text{Buffer}}-[O_2]_{\text{Buffer}}$ plane.

[81] Eddy, S. What is dynamic programming?. Nat Biotechnol 22, 909–910 (2004).

R2.8: Minor comments:

- Lines 140-142: Can the authors comment on the mechanism of these 26Hz oscillations? Are these mediated by the kinetics of inhibition/IPSCs in the network?

A2.8: Yes, these oscillations are mediated by inhibitory synaptic kinetics. We have added comments on the mechanism to the main text. Extra analysis performed during the revisions revealed that the originally stated ~26 Hz was a noisy estimate and using a longer time window gives a better estimate of ~31 Hz. We have also corrected this in the main text.

Results (p6):

This network rhythm emerges through the interaction between excitatory and inhibitory neurons. Increasing the time constant of inhibitory synapses slows down the response of the inhibitory neurons to incoming input. This delay allows excitatory neurons to increase their firing rate transiently before inhibition reduces it again, which drives a roughly oscillatory modulation of the network firing rate. The peak frequency of ~31 Hz increases with decreasing τ_{inh} in the vicinity of our nominal parameter set (Supp. Fig. S1).

Supplementary Figure S1: Dependence of AI state network oscillatory frequency on inhibitory time constant. Curves show log(power spectral density) for a series of values of τ_{inh} . Red line joins the spectral peaks to illustrate the relationship between the peak frequency and τ_{inh} .

R2.9: - *I would prefer that code be made available through a publicly accessible repository, such as GitHub, rather than by request on the authors'*

A2.9: We have put the code on GitHub: <https://github.com/brain-modelling-group/neuro-metabolic-model>

REVIEWERS' COMMENTS

Reviewer #1 (Remarks to the Author):

The authors addressed all my comments. However, I have some questions.

- 1) What is the biophysical meaning of [K⁺]buffer and [O₂]buffer?
- 2) From Figure S2e, it donot look like a seizure, how did the authors define a seizure?
- 3) How does the parameters inferred from the infant? How is the good and poor recovery defined?

Reviewer #2 (Remarks to the Author):

I have reviewed the response and revised manuscript. The author's have performed a thorough revision that sufficiently addresses by prior concerns. I have no further objections to the publication of this manuscript.

We thank the reviewers for their comments. Our responses (black) and changes to the text (red) are interleaved with the comments (italic).

Reviewer #1: *The authors addressed all my comments. However, I have some questions.*

We thank the reviewer for accepting the revisions and for their constructive comments.

RI.1: *What is the biophysical meaning of $[K^+]_{\text{buffer}}$ and $[O_2]_{\text{buffer}}$?*

A1.1: We now clarify this in methods:

Methods (p14):

The supply of O_2 locally to a neuron is modeled as diffusion from the cerebral circulation, assumed to be a metabolic reserve with fixed concentration $[O_2]_{\text{Buffer}}$ [16].

Methods (p14):

The parameter $[K^+]_{\text{Buffer}}$ is an effective value describing the collective buffering capacity of various sources of potassium, particularly crucial during metabolically-challenged states.

Methods (p15):

All these sources, along with the vasculature, contribute to the buffering capacity parameterized by $[K^+]_{\text{Buffer}}$ in our model.

RI.2: *From Figure S2e, it donot look like a seizure, how did the authors define a seizure?*

A1.2: We now clarify how we defined seizures, and that Figure S2e shows only two neurons giving only a limited window into the seizure dynamics:

Methods (p19):

In particular, a seizure in this context is further defined by the sudden onset of high firing with marked synchronicity among neural spikes and sustained fluctuations in the firing rate time-series.

Supplementary information (p2):

Supplementary Figure S2: e) Seizure, marked by synchronicity between the selected excitatory and inhibitory neurons. This panel illustrates the dynamics of two neurons during a seizure state, but it should be noted that a full seizure event involves widespread synchronicity across an ensemble of neurons, which is better captured at the network level (see Fig. 6 in the main text).

RI.3: *How does the parameters inferred from the infant? How is the good and poor recovery defined?*

A1.3: We have added clarifications to the Methods text on parameter inference, and the definition of good and poor recovery:

Methods (p19):

To test the validity of our model, we estimated model parameters that yield dynamics consistent with infant EEG during recovery from hypoxia. To do this, we analyzed scalp EEG recordings from 17 infants admitted to the tertiary level neonatal intensive care unit (NICU) in Helsinki University Central Hospital due to perinatal asphyxia (see [8] for details on this pre-existing dataset). The use of the retrospectively collected, archived patient data was approved by the Ethics Committee of the Hospital for Children and Adolescents, Helsinki University Central Hospital. Neurodevelopmental outcome categories were identified previously [8], such that good recovery (10 infants) denotes either normal development or only mild neuromuscular disorders as assessed at their last visit to a routine neonatal outpatient clinic (age 12–39 months), while poor recovery (7 infants) denotes either death, severe neuromuscular disorders (severe injury), or a thalamic lesion.

Methods (p20):

We then estimated each infant's trajectory in the $[K^+]_{\text{Buffer}}-[O_2]_{\text{Buffer}}$ plane. Our approach uses a dynamic programming framework to minimize differences between EEG features captured from infants and those predicted by our model. We summarized the i^{th} window from the j^{th} epoch with a vector of these six features (D_i^j).

Reviewer #2: *I have reviewed the response and revised manuscript. The author's have performed a thorough revision that sufficiently addresses by prior concerns. I have no further objections to the publication of this manuscript.*

We thank the reviewer for accepting the revisions.